# The classical pathway triggers pathogenic complement activation in membranous nephropathy

Larissa Seifert[1,2], Gunther Zahner[1,2], Catherine Meyer-Schwesinger[2,3], Naemi Hickstein[1,2], Silke Dehde[1,2], Sonia Wulf [4], Sarah M. S. Köllner[1,2], Renke Lucas[1,2], Dominik Kylies[1,2], Sarah Froembling[2,3], Stephanie Zielinski[2,3], Oliver Kretz[1,2], Anna Borodovsky[5], Sergey Biniaminov[6], Yanyan Wang[7], Hong Cheng[7], Friedrich Koch-Nolte[8], Peter F. Zipfel[9,10], Helmut Hopfer[11], Victor G. Puelles [1,2,12,13], Ulf Panzer[1,2,14], Tobias B. Huber [1,2], Thorsten Wiech [2,4,15] & Nicola M. Tomas [1,2,15] ✉

Membranous nephropathy (MN) is an antibody-mediated autoimmune disease characterized by glomerular immune complexes containing complement components. However, both the initiation pathways and the pathogenic significance of complement activation in MN are poorly understood. Here, we show that components from all three complement pathways (alternative, classical and lectin) are found in renal biopsies from patients with MN. Proximity ligation assays to directly visualize complement assembly in the tissue reveal dominant activation via the classical pathway, with a close correlation to the degree of glomerular C1q-binding IgG subclasses. In an antigen-specific autoimmune mouse model of MN, glomerular damage and proteinuria are reduced in complement-deficient mice compared with wild-type littermates. Severe disease with progressive ascites, accompanied by extensive loss of the integral podocyte slit diaphragm proteins, nephrin and neph1, only occur in wild-type animals. Finally, targeted silencing of C3 using RNA interference after the onset of proteinuria significantly attenuates disease. Our study shows that, in MN, complement is primarily activated via the classical pathway and targeting complement components such as C3 may represent a promising therapeutic strategy.

[1]III. Department of Medicine, University Medical Center Hamburg-Eppendorf, Hamburg, Germany. [2]Hamburg Center for Kidney Health (HCKH), University Medical Center Hamburg-Eppendorf, Hamburg, Germany. [3]Institute of Cellular and Integrative Physiology, University Medical Center Hamburg-Eppendorf, Hamburg, Germany. [4]Institute of Pathology, Nephropathology Section, University Medical Center Hamburg-Eppendorf, Hamburg, Germany. [5]Alnylam Pharmaceuticals, Cambridge, USA. [6]HS Analysis GmbH, Karlsruhe, Germany. [7]Division of Nephrology, Beijing Anzhen Hospital, Capital Medical University, Beijing, China. [8]Institute of Immunology, University Medical Center Hamburg-Eppendorf, Hamburg, Germany. [9]Department of Infection Biology, Leibniz Institute for Natural Product Research and Infection Biology, Hans Knöll Institute, Jena, Germany. [10]Institute of Microbiology, Friedrich Schiller University, Jena, Germany. [11]Department of Medical Genetics and Pathology, University Hospital Basel, University of Basel, Basel, Switzerland. [12]Department of Clinical Medicine, Aarhus University, Aarhus, Denmark. [13]Department of Pathology, Aarhus University Hospital, Aarhus, Denmark. [14]Hamburg Center for Translational Immunology (HCTI), University Medical Center Hamburg-Eppendorf, Hamburg, Germany. [15]These authors jointly supervised this work: Thorsten Wiech, Nicola M. Tomas. ✉e-mail: n.tomas@uke.de

Membranous nephropathy (MN) is an antibody-mediated autoimmune disease and a major cause of nephrotic syndrome in adult patients[1]. Morphologically, granular deposition of immunoglobulins (IgG) and complement components can be detected by immunofluorescence, suggesting a role of both autoantibodies and the complement system in the pathogenesis of MN[2,3]. To date, autoantibodies against several antigens have been identified in patients with MN[4–10], with the phospholipase A2 receptor 1 (PLA2R1) and thrombospondin type-1 domain containing protein 7A (THSD7A) accounting for around 75% of MN cases[4,5].

The complement system can be initiated via three different pathways, classical, lectin, and alternative (Supplementary Fig. 1a). The classical pathway is activated by binding of C1q to antigen-antibody complexes, leading to cleavage of C2 and C4, which enables assembly of the classical/lectin C3 convertase C4bC2b. The lectin pathway is initiated by pattern recognition molecules such as mannan-binding lectin (MBL), which bind to MBL-associated serine proteases (MASPs). This also induces cleavage of C2 and C4 and thus formation of the classical/lectin C3 convertase C4bC2b. The alternative pathway is constitutively activated by slow hydrolysis of C3, which, in the presence of complement factors B and D, leads to the formation of the alternative C3 convertase C3bBb. Both the classical/lectin and the alternative C3 convertases cleave fluid phase C3, and subsequent progression results in assembly of the classical and alternative C5 convertases, resulting in cleavage of C5 and formation of the membrane attack complex, C5b-9. Given the setting of an antibody-mediated disease, it is reasonable to assume that activation of the complement system in MN is initiated by antigen-antibody complexes, i.e., via the classical pathway. However, autoantibodies against PLA2R1 and THSD7A are predominantly of the IgG4 subclass, the IgG subclass with the least C1q-binding capacity and thus limited (or no) ability to activate the complement system via the classical pathway[11]. In line with this, in an early report, C1q was found to be absent in immune deposits of MN patients[12]. By contrast, other studies found variable frequencies of C1q deposition ranging from 17 to 80% of MN cases[3,13–15], and the presence of complement-activating IgG subclasses both in serum and glomeruli from MN patients[16–19]. Less controversy exists on the glomerular deposition of the central complement component C3 and the membrane attack complex, C5b-9[20,21], which indicates complement activation, but does not allow conclusions on the initiating pathway(s). Interestingly, C4d, a cleavage product of C4b, is present in the majority of MN biopsies[22,23], hinting to a role of the classical pathway, the lectin pathway, or both. Indeed, MBL has been detected in glomeruli of patients with MN, and MBL levels were linked to the clinical outcome of affected patients, yet with contradictory results[23–25]. More recently, anti-PLA2R1 IgG4 with distinct glycosylation patterns was shown to directly bind MBL in vitro, thus activating the complement system via the lectin pathway[26]. However, MN can also occur in patients with a genetic MBL deficiency, suggesting a role of other complement pathways or an alternative route of lectin pathway activation[27].

While the presence of complement components at the site of glomerular injury in biopsies from patients with MN is beyond dispute, the pathogenic significance of these complement deposits is less clear. Investigations related to the pathogenicity of complement activation in MN have been hampered by the lack of appropriate animal models. The main requirement for such an animal model would be the accurate reproduction of the disease including the main clinical (i.e., proteinuria) and morphological (i.e., glomerular IgG and complement deposition) characteristics as well as key pathophysiological features such as involvement of autoantibodies against one of the known target antigens.

Taken together, the initiation pathways and the pathogenic relevance of complement activation in MN are incompletely understood. Here, we analyze complement activation in an exemplary cohort of patients with PLA2R1- and THSD7A-associated MN and provide evidence for a dominant role of the classical pathway. Based on this finding, we further investigate the pathogenic significance of complement activation in a newly established, antigen-specific, autoimmune mouse model of MN. We demonstrate that in this model, both genetic C3 deficiency and therapeutic targeting of C3 attenuate glomerular damage and disease severity. Our results support a pathogenic role of the complement system in MN, but also suggest a role of complement-independent injury pathways.

## Results

### Complement components from all three complement pathways are present in biopsies from patients with MN

Complement components such as C3 and C5b-9 are usually detectable in kidney biopsies from patients with MN[3,20,21,28,29], but the initiation pathways upstream of C3 are incompletely understood. We initially evaluated complement deposition in a pilot cohort of three patients with PLA2R1- and two patients with THSD7A-associated MN by staining for key components of the three complement initiation pathways (Supplementary Fig. 1a, b) in paraffin-embedded kidney biopsy samples. We found strong positivity and co-localization of C3b and CFB as well as C4b and C2 in 2/5 and 5/5 investigated cases, respectively (Fig. 1a, b). Complement components located to the outer aspect of collagen type IV, a constituent of the glomerular basement membrane (GBM), indicating deposition in the subepithelial space near the podocyte cell membrane. We next stained for MBL and C1q, the complement components that initiate the lectin and classical pathways, respectively (Supplementary Fig. 1a). MBL stained positive and co-localized with IgG in 2/5 MN biopsies (Fig. 1c), supporting the recent finding that an interaction between IgG and MBL may directly activate the lectin pathway of complement in a subset of cases[26]. C1q was found to be positive and in co-localization with IgG in the subepithelial space in all investigated MN cases using two different anti-C1q antibodies (Fig. 1d, Supplementary Figs. 2 and 3). By contrast, C1q was negative at the glomerular filtration barrier in cases of primary focal segmental glomerulosclerosis (FSGS), minimal change disease (MCD), diabetic nephropathy (DN), and amyloidosis (Supplementary Figs. 3 and 4), indicating that C1q fixation is a primary event in MN and not a consequence of glomerular filtration barrier damage and proteinuria per se. In four control biopsy samples (time point zero biopsies from renal transplant recipients), C3b, CFB, C4b, C2, C1q, IgG, and MBL were all negative (Fig. 1a–d).

As studies using immunofluorescence on cryo-conserved kidney tissue (as opposed to the paraffin-embedded tissue used for the stainings shown in Fig. 1a–d) found much less glomerular C1q positivity in MN[3,12–14], we hypothesized that C1q is masked in immune complexes by IgG and other complement components[30] and investigated frozen biopsy sections from MN patients and controls (Supplementary Fig. 1d). Indeed, while C1q was positive in cases of class V membranous lupus nephritis (n = 2), it was found to be negative in cases of PLA2R1-associated MN (n = 6) and minimal change disease (n = 2) when conventional immunofluorescence was done on frozen sections of kidney biopsies without tissue preprocessing (Fig. 1e and Supplementary Fig. 5). However, C1q became positive in all investigated cryo-conserved samples from patients with PLA2R1-associated MN, but not in samples from patients with minimal change disease, when an antigen retrieval with methanol and trypsin was applied (Fig. 1e and Supplementary Fig. 6). Together, these experiments show prominent glomerular deposition of complement components including C1q, the initiator of the classical pathway of complement, in kidney biopsies from patients with PLA2R1- and THSD7A-associated MN.

### The classical pathway is dominantly activated in patients with MN

Considering our observation that components from all three complement pathways were found in MN biopsies using

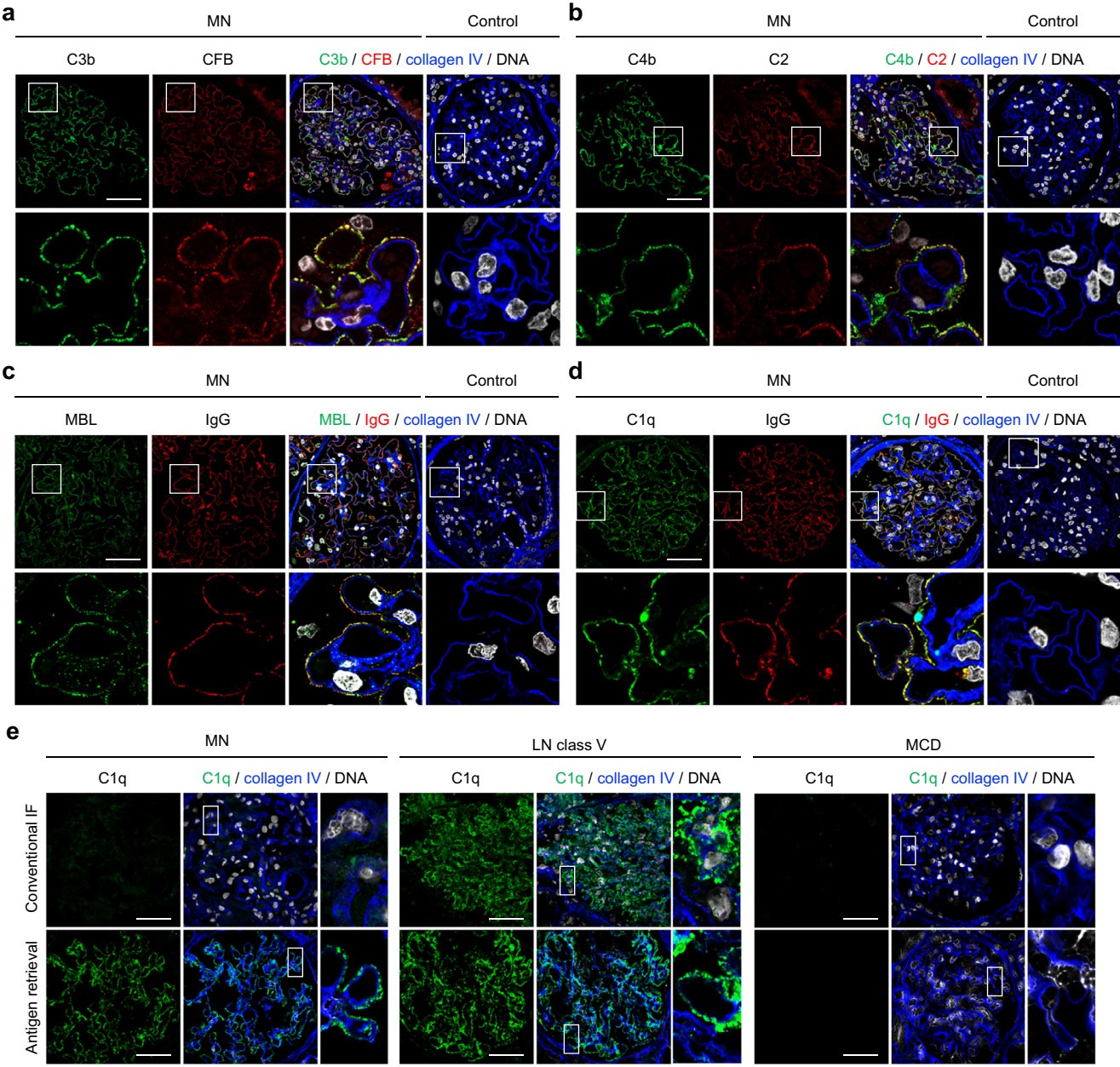

**Fig. 1 | Complement components from all three complement activation pathways can be detected in biopsies from MN patients. a–d** Representative immunofluorescence (IF) stainings (paraffin-embedded tissue) for C3b and CFB (**a**), C4b and C2 (**b**), MBL and IgG (**c**), and C1q and IgG (**d**) in co-localization with the glomerular basement membrane constituent collagen IV in biopsies from MN patients (*n* = 5) and controls (*n* = 4). The lower panels represent 5-fold enlargements of the boxed areas in the upper panels. Bars 50 μm. **e** Representative IF stainings (frozen tissue) for C1q in co-localization with collagen IV in biopsies from patients with MN (*n* = 6), patients with lupus nephritis (LN) class V (*n* = 2), and patients with minimal change disease (MCD, *n* = 2) using conventional indirect IF (upper panels) and IF after antigen retrieval with methanol and trypsin (lower panels). Panels on the right represent 5-fold enlargements of the boxed areas in the left panels. Bars 50 μm.

immunofluorescence, we investigated complement activation using proximity ligation assays. This technique allows the detection of proteins in direct proximity, in this case the classical/lectin C3 convertase (involving C4b and C2b) and the alternative C3 convertase (involving C3b and Bb, a fragment of CFB), indicating convertase formation in the tissue and thus complement activity[31]. We analyzed kidney biopsies from 39 MN patients (29 with PLA2R1- and 10 with THSD7A-associated MN, Supplementary Table 1), as well as six control biopsies, which were biopsies from kidney allografts at the time of transplantation (Supplementary Fig. 1c). We found glomerular signals for the alternative C3 convertase C3bBb in two thirds (26 of 39) and for the classical/lectin C3 convertase C4bC2b in all (39 of 39) of the investigated MN cases (Fig. 2a, b), while the convertases were absent in the allograft control

biopsies (Supplementary Fig. 7a). We next analyzed the upstream complement initiation that led to formation of C4bC2b. The proximity ligation assay for IgGC1q (indicating classical pathway activation) was positive in 34 out of 39 MN biopsy samples (Fig. 2b, c). MBL deposition was analyzed by immunofluorescence and by an IgGMBL proximity ligation assay and was detected in 21 out of 35 still available biopsy samples (Fig. 2b, c), indicating lectin pathway activity near deposited IgG. Both IgGC1q and IgGMBL were negative in control biopsies (Supplementary Fig. 7b). When quantifying proximity ligation assay signals per glomerular area, IgGC1q showed the highest signal intensity, followed by the classical/lectin C3 convertase C4bC2b, the alternative C3 convertase C3bBb, and IgGMBL (Fig. 2d). Together, these experiments indicate activation of all three complement pathways with

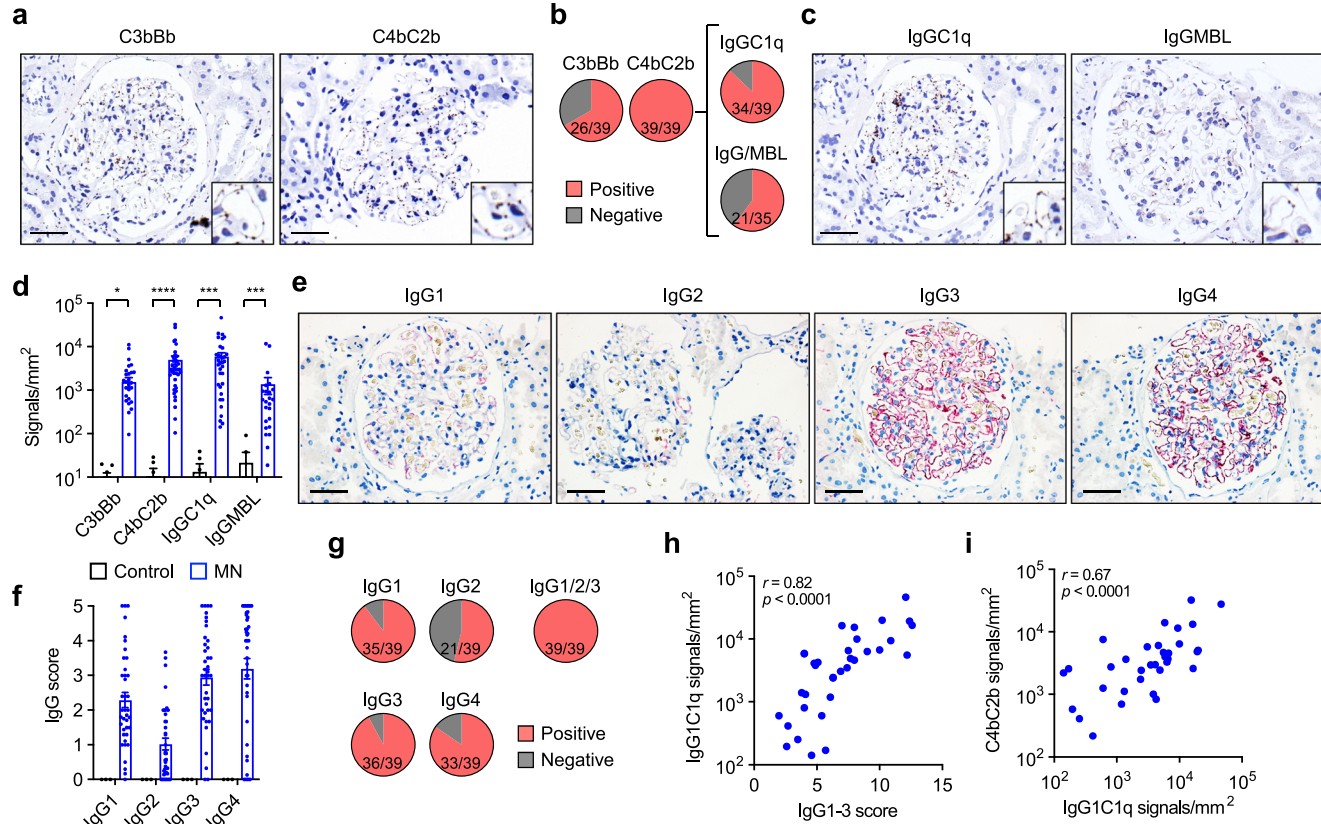

**Fig. 2 | Complement is dominantly activated via the classical pathway in patients with MN. a** Representative images of proximity ligation assays showing the alternative convertase C3bBb (left) and the classical/lectin convertase C4bC2b (right) performed on biopsies from patients with MN ($n = 39$). Bars 50 µm. **b** Detection of C3bBb, C4bC2b and IgGC1q using proximity ligation assays as well as of MBL/IgG in immunofluorescence and proximity ligation assay. **c** Representative images of proximity ligation assays showing the assembly of IgGC1q (left) and IgGMBL (right) performed on biopsies from patients with MN ($n = 39$ and $n = 26$ for IgGC1q and IgGMBL, respectively). Bars 50 µm. **d** Quantitative analysis of proximity ligation assay signals in biopsies from patients with MN and time point zero biopsies from renal transplant recipients. Data are presented as mean and SEM.

C3bBb, *$p = 0.0442$; C4bC2b, ****$p < 0.0001$; IgGC1q, ***$p = 0.0005$; IgGMBL, ***$p = 0.0003$ (two-tailed Mann–Whitney test). **e** Representative immunohisto-chemical stainings for the IgG subclasses IgG1, IgG2, IgG3, and IgG4 in a biopsy sample from a patient with MN ($n = 39$). Bars 50 µm. **f** Quantitative analysis (histo-logical score) of IgG subclass signals in 39 biopsies from patients with MN and 3 time point zero biopsies from renal transplant recipients. Data are presented as mean and SEM. **g** Detection of IgG1, IgG2, IgG3, and IgG4 as well as at least one of the C1q-binding subclasses IgG1, IgG2, or IgG3 in biopsies from patients with MN. **h, i** Correlation of the cumulative IgG1-3 score with IgGC1q proximity ligation assay signals (**h**) and IgGC1q signals with the C4bC2b signals (**i**). Spearman's $r$ correlation coefficient (two-tailed).

predominance of the classical pathway in patients with PLA2R1- and THSD7A-associated MN.

## Glomerular deposition of C1q-binding IgG subclasses is found in all patients with MN

As non-C1q-binding IgG4 was reported to be the dominant IgG sub-class in patients with primary MN[4,5], but we found prominent com-plement activation via the classical pathway, we next mapped the distribution of IgG subclasses in biopsy samples from our cohort of PLA2R1- and THSD7A-associated MN cases using immunohistochem-istry (Fig. 2e). When scoring the amount of deposited IgG subclasses semi-quantitatively, we found the strongest positivity for IgG4, fol-lowed by IgG3, IgG1, and IgG2 (Fig. 2f). Notably, at least one of the complement-activating IgG subclasses IgG1, IgG2, or IgG3 was positive in all 39 investigated biopsies, with IgG3, the human IgG subclass with the highest C1q-binding capacity[11], being the most prevalent of all subclasses (Fig. 2g). The cumulative score of the C1q-binding IgG subclasses IgG1-3 strongly correlated with the IgGC1q proximity liga-tion assay signals (Fig. 2h) and the IgGC1q signals strongly correlated with the C4bC2b signals (Fig. 2i), suggesting classical pathway activa-tion by C1q-binding IgG. In line with this, the individual IgG1, IgG2, and IgG3 scores, but not the IgG4 score, also correlated with IgGC1q signals (Supplementary Fig. 8a). Interestingly, glomerular IgG1 and

IgG4 showed a close correlation with IgGMBL proximity ligation assay signals and IgGMBL signals correlated with C4bC2b signals (Supple-mentary Fig. 8b, c), suggesting a relevant contribution of the lectin pathway to classical/lectin C3 convertase activation in a subset of cases. In conclusion, our observations indicate that C1q-binding IgG subclasses are important drivers of complement activation via the classical/lectin C3 convertase C4bC2b in patients with MN.

## Immunization with THSD7A fragments induces MN with nephrotic syndrome in mice

To reproduce the autoimmune pathogenesis of MN, we actively immunized mice with the murine domains of THSD7A that are most frequently recognized by patient autoantibodies (Fig. 3a)[17]. The frag-ments were designed, expressed in HEK293 cells, and purified (Fig. 3b). We chose THSD7A as the target antigen for our model because it is, in contrast to PLA2R1, highly expressed in rodent podocytes[32], and we chose BALB/c mice as this strain favors a Th2 cytokine production with a strong humoral response and was susceptible to develop MN after passive antibody transfer in previous studies[32–35]. Mice received four immunizations with a mix of the four THSD7A fragments (THSD7A-immunized group, $n = 10$) or with PBS (control group, $n = 5$), both in combination with Freund's adjuvant, over 7 weeks and were monitored for a total of 20 weeks for the development of anti-THSD7A

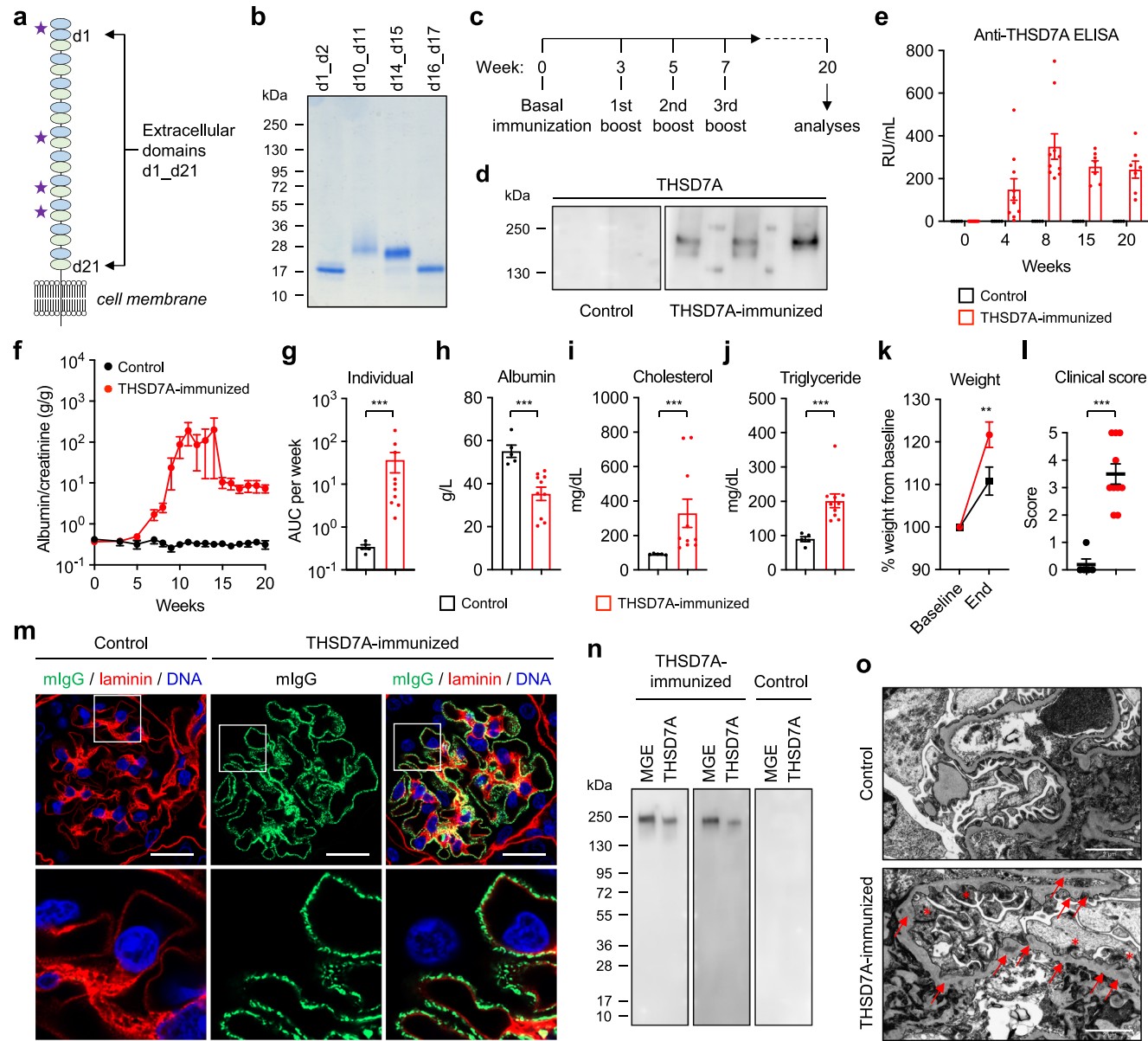

**Fig. 3 | Active immunization with THSD7A fragments induces membranous nephropathy in mice. a** Molecular architecture of THSD7A with 21 extracellular domains. Asterisks mark the four regions that were used for active immunization. **b** Coomassie-blue staining of the four THSD7A fragments that were used for immunization (representative gel from at least 10 experiments). **c** Immunization scheme. Mice were immunized with PBS (further referred to as control, n = 5) or THSD7A fragments (further referred to as THSD7A-immunized, n = 10), both in combination with Freund's adjuvant, in two independent experiments. **d** Western blot analyses of mouse sera (from 2 controls and 4 THSD7A-immunized animals) taken 4 weeks after immunization on recombinant THSD7A. **e, f** Anti-THSD7A antibody titers as measured by ELISA (**e**) and albuminuria as measured by albumin-to-creatinine ratio (**f**) over time. **g** Individual albuminuria as measured by the area under the curve (AUC) per week. ***p = 0.0007 (two-tailed Mann–Whitney test). **h–j** Serum values for albumin (**h**, ***p = 0.0007), cholesterol (**i**, ***p = 0.0007) and

triglyceride (**j**, ***p = 0.0007) (two-tailed Mann–Whitney test). **k** Weight change in percent. **p = 0.0079 (two-way ANOVA with Bonferroni correction for multiple comparisons). **l** Nephrotic syndrome clinical score. ***p = 0.0003 (two-tailed Mann–Whitney test). In panels **e–l**, data are presented as mean and SEM and derived from n = 5 controls and n = 10 THSD7A-immunized animals. **m** Representative immunofluorescence staining for mouse IgG (mIgG) in co-localization with laminin in control (n = 5) and THSD7A-immunized (n = 10) mice. Bars 20 μm. Lower panels represent 3.5-fold enlargements of the boxed areas in the upper panels. **n** Western blot analysis of antibodies eluted from frozen kidney sections of experimental mice on mouse glomerular extracts (MGE) and recombinant THSD7A (n = 1 experiment). **o** Representative electron microscopic analysis of the glomerular filtration barrier in control and THSD7A-immunized mice (n = 5 per group). Red arrows point at subepithelial electron-dense deposits. Red asterisks mark effaced podocyte foot processes. Bars 2 μm.

autoantibodies and proteinuria (Fig. 3c). Four weeks after the first immunization, anti-THSD7A antibodies could be detected using Western blotting in the THSD7A-immunized mice, but were absent in controls (Fig. 3d). Anti-THSD7A titers were measured using an ELISA (Supplementary Fig. 9a) before immunization and then 4, 8, 15, and 20 weeks after immunization. By week 8, all THSD7A-immunized mice had developed relevant titers of anti-THSD7A antibodies (Fig. 3e). Starting 5 weeks after immunization, proteinuria, as measured by

albumin-to-creatinine ratio, started to increase in THSD7A-immunized mice compared to their baseline values that were measured before immunization (Fig. 3f). Interestingly, 3/10 mice in the THSD7A-immunized group developed severe proteinuria with progressive ascites and marked weight gain, which required an early termination of the experiment for the affected animals (Supplementary Fig. 9b). As albuminuria varied inter- and intra-individually (Supplementary Fig. 9c), we considered the albumin-to-creatinine area under the curve

(AUC) per week as a valid measure of the individual level of proteinuria (rather than a single time point during the observation period). THSD7A-immunized mice had significantly higher overall albuminuria than control mice (Fig. 3g). Proteinuria in THSD7A-immunized mice was accompanied by significantly lower serum albumin (Fig. 3h), higher serum cholesterol (Fig. 3i) and higher serum triglycerides (Fig. 3j) when compared to controls at the end of the observation period. THSD7A-immunized mice also showed a significantly higher increase in body weight compared to control mice (Fig. 3k), suggesting fluid accumulation as a consequence of prominent albuminuria. When calculating a combined nephrotic syndrome clinical score, which involves the parameters proteinuria, hypoalbuminemia, hyperlipidemia, and weight gain, THSD7A-immunized animals had significantly higher values than controls (Fig. 3l). In contrast to BALB/c mice, C57BL/6 mice did not develop significant proteinuria after immunization (Supplementary Fig. 10). Together, these results show the development of a nephrotic syndrome in wild-type BALB/c mice upon immunization with THSD7A.

### Immunization with THSD7A induces the histological and ultrastructural changes of MN in mice

We next investigated the histological changes after active immunization with THSD7A. Light microscopic evaluation demonstrated a slight thickening of the GBM in PAS stain and numerous non-argyrophilic pinholes in the Jones' methenamine silver stain leading to a vacuolated appearance of the GBM in THSD7A-immunized mice, but not in controls, both of which are typical light microscopic signs of MN (Supplementary Fig. 11a, b). In immunofluorescence, glomeruli of THSD7A-immunized mice were strongly positive for mouse IgG (mIgG) (Fig. 3m). It was found to be granular and located on the outer, subepithelial aspect of the GBM, which is considered a histological hallmark of MN. Moreover, glomerular mIgG deposits strongly co-localized with THSD7A in granular immune complexes, suggesting specificity of the deposited antibodies for THSD7A (Supplementary Fig. 11c). No mIgG deposition was observed at the glomerular filtration barrier in control mice (Fig. 3m and Supplementary Fig. 11c). Similar to human MN cases, we did not detect substantial renal cellular infiltrates of T cells, macrophages or dendritic cells in THSD7A-immunized mice compared with controls, as indicated by absence of CD3- and F4/80-positive cells (Supplementary Fig. 11d). IgG eluted from THSD7A-immunized mice, but not from control mice, recognized a protein of 250-kDa present in mouse glomerular extracts (MGE) and also recombinant THSD7A (Fig. 3n), demonstrating specificity of the glomerular IgG for THSD7A. In electron microscopic analyses, THSD7A-immunized mice showed electron-dense deposits in a subepithelial and intramembranous localization, intervening projections of extracellular matrix, forming spikes or encasing the deposits, as well as extensive podocyte foot process effacement (Fig. 3o).

To further control for the specificity of our findings in THSD7A-immunized animals, we immunized wild-type BALB/c mice with ovalbumin as well as Thsd7a$^{-/-}$ mice with THSD7A. Ovalbumin-immunized animals developed anti-ovalbumin antibodies, but no albuminuria and no glomerular mIgG and complement deposition (Supplementary Fig. 12). Thsd7a$^{-/-}$ mice developed high levels of anti-THSD7A antibodies, but, in contrast to their wild-type (WT) littermates, no albuminuria, no glomerular mIgG, and no complement deposition (Supplementary Fig. 13). In conclusion, these experiments demonstrate the successful establishment of an antigen-specific autoimmune mouse model of MN involving one of the known target antigens.

### mIgG subclasses and complement in mice after immunization with THSD7A

Considering that MN patients have circulating autoantibodies of different IgG subclasses[18] and biopsies from patients with PLA2R1- and THSD7A-associated MN contain glomerular deposits of both complement-activating and non-activating IgG subclasses (Fig. 2f), we investigated serum levels and glomerular deposition of mIgG subclasses in the model. Murine IgG1, which resembles human IgG4 in lacking the capacity to bind C1q and thus to activate complement via the classical pathway[36], was found to be the most abundant mIgG subclass in serum and glomeruli of THSD7A-immunized mice, followed by mIgG2a, mIgG2b, and mIgG3 (Fig. 4a–c). The classical pathway initiator C1q was positive in all THSD7A-immunized mice and co-localized with mIgG, suggesting classical pathway activation by locally bound anti-THSD7A autoantibodies (Fig. 4d). In addition to C1q, C4d as a marker of the classical or lectin pathway, CFB as the main activator of the alternative pathway, CFH as an inhibitor of the alternative pathway, the central complement component C3, and the membrane attack complex C5b-9 all stained positive in THSD7A-immunized mice and negative in controls (Fig. 4e and Supplementary Fig. 14a). No significant mIgG and complement deposition was found in organs other than the kidney, arguing against non-specific systemic complement activation resulting from THSD7A immunization (Supplementary Fig. 15). As glomerular injury and proteinuria have been associated with formation and membrane insertion of C5b-9[37], we scored the amount of deposited C5b-9 semi-quantitatively. THSD7A-immunized mice differed in the amount of deposited C5b-9 (Fig. 4f and Supplementary Fig. 14b). While there was no significant correlation between glomerular non-complement activating mIgG1 signals and C5b-9 deposition, complement-activating mIgG2a, 2b, 3 as well as the cumulative mIgG2-3 signals strongly correlated with C5b-9 deposition (Fig. 4g), suggesting complement activation by mIgG subclasses 2a, 2b, and 3. C5b-9 deposition in turn strongly correlated with proteinuria, suggesting that C5b-9-mediated damage influences the degree of proteinuria (Fig. 4h). In summary, these results demonstrate classical and alternative pathway activation and suggest an involvement of both IgG and complement activation in the pathogenesis of experimental autoimmune MN.

### Genetic complement C3 deficiency ameliorates experimental autoimmune MN

Next, we investigated the pathogenic role of the complement system in MN using genetically C3-deficient BALB/c mice (C3$^{-/-}$ mice) (Supplementary Fig. 16a, b). We applied the previously established immunization protocol in C3$^{-/-}$ mice (n = 20) and in WT littermate controls (n = 19). A mixed group of C3$^{-/-}$ mice and wild-type littermates that was immunized with PBS served as the baseline control (n = 7). After immunization with THSD7A, C3$^{-/-}$ mice and WT littermates developed comparable levels of anti-THSD7A autoantibodies (Fig. 5a). While 4/19 mice from the WT littermate group developed severe fluid retention and had to be euthanized before week 20, C3$^{-/-}$ mice were protected from such disease exacerbation (Fig. 5b). Moreover, proteinuria developed earlier and was higher in WT littermates than in C3$^{-/-}$ mice both when plotted over time and as individual proteinuria as measured by the AUC per week (Fig. 5c, d), indicating an important role of activated complement in the mediation of podocyte damage and proteinuria in experimental autoimmune MN. The reduced proteinuria in C3$^{-/-}$ mice was accompanied by a less severe nephrotic syndrome (Fig. 5e and Supplementary Fig. 16c). All mice immunized with THSD7A showed strong granular deposition of mIgG at the capillary walls in immunofluorescence (Supplementary Fig. 16d). Similar to WT mice, C3$^{-/-}$ mice and their WT littermates had a strong mIgG1 response, followed by mIgG2a, mIgG2b and mIgG3 (Fig. 5f and Supplementary Fig. 16e). Complement C1q and C4d were positive in all investigated THSD7A-immunized mice, while C3, C5b-9, CFB, and CFH were positive in WT littermates, but negative or barely detectable in C3$^{-/-}$ mice (Supplementary Fig. 16f), indicating that the complement cascade is also initiated in C3$^{-/-}$ mice, but is terminated at the level of C3, preventing both the formation of C5b-9 and the activation of the alternative pathway through hydrolysis of C3. C5b-9 deposition varied in

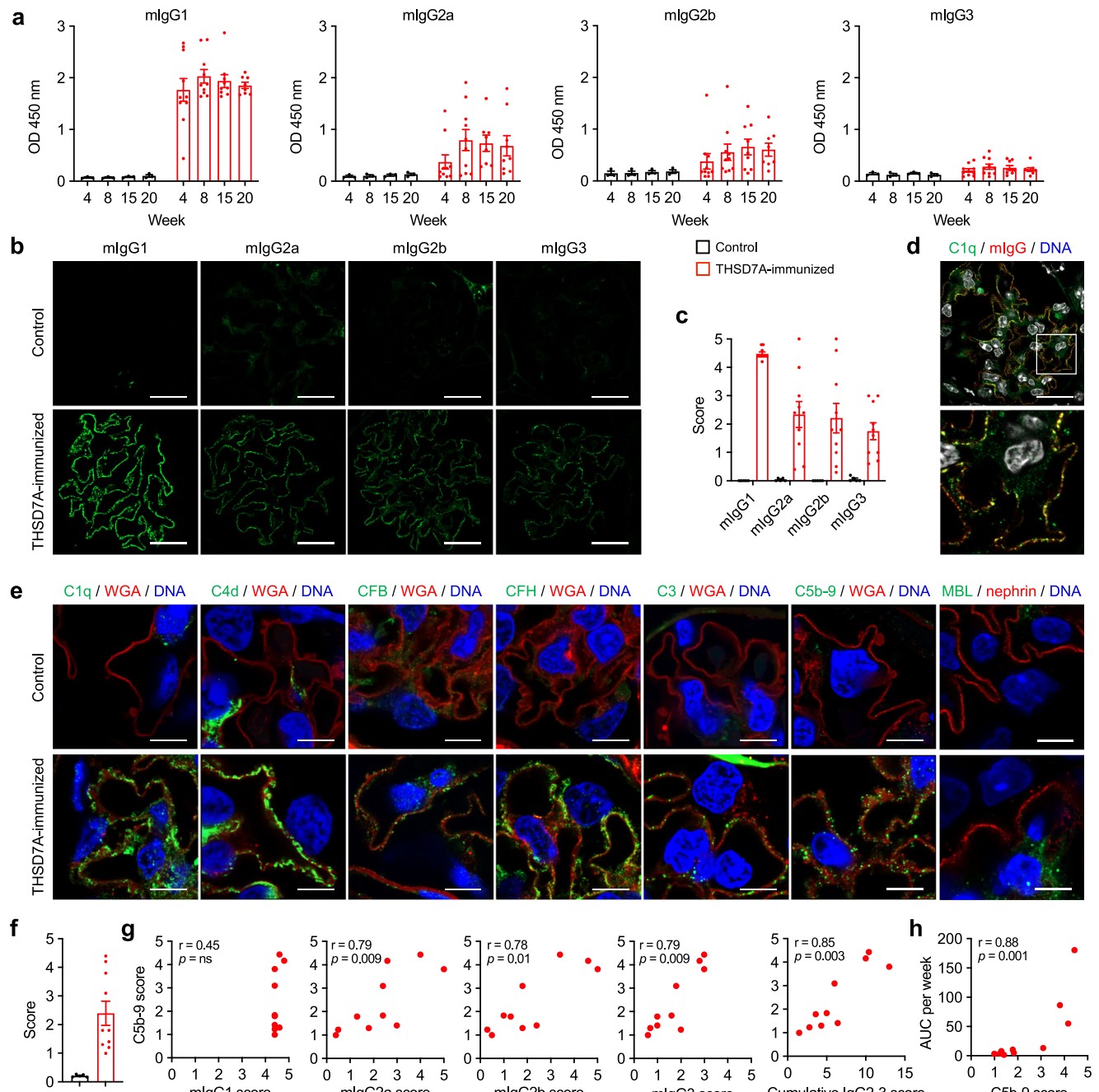

**Fig. 4 | Analysis of IgG subclasses and complement deposition in experimental autoimmune MN. a** mIgG subclass-specific anti-THSD7A antibody levels over time as measured by ELISA in controls (n = 3) and THSD7A-immunized mice (n = 10). Data are presented as mean and SEM. **b** Representative immunofluorescence stainings for mIgG1, mIgG2a, mIgG2b, and mIgG3 in controls (n = 5) and THSD7A-immunized mice (n = 10). Bars 20 μm. **c** Quantitative analysis (histological score) of glomerular IgG subclass deposition in control and THSD7A-immunized mice. Data are presented as mean and SEM. **d** Representative immunofluorescence staining of complement C1q in co-localization with mouse IgG (mIgG) in THSD7A-immunized mice (n = 4). Bar 20 μm. The lower image is an enlargement of the boxed area in the upper image. **e** Representative immunofluorescence stainings for complement C1q,

C4d, CFB, CFH, C3, and C5b-9 in co-localization with wheat germ agglutinin (WGA) and MBL in co-localization with nephrin in control (n = 5) and THSD7A-immunized mice (n = 10). Bars 4 μm. **f** Quantitative analysis (histological score) of glomerular C5b-9 deposition in control (n = 5) and THSD7A-immunized mice (n = 10). Data are presented as mean and SEM. **g** Correlation analyses of the mIgG1, mIgG2a, mIgG2b and mIgG3 scores with the C5b-9 score and correlation of the cumulative mIgG2a, mIgG2b, and mIgG3 (referred to as mIgG2-3) score with the C5b-9 score in THSD7A-immunized mice (n = 10). ns, not significant. Spearman's r correlation coefficient (two-tailed). **h** Correlation analysis of C5b-9 scores with individual albuminuria values as measured by the area under the curve (AUC) per week in THSD7A-immunized mice (n = 10). Spearman's r correlation coefficient (two-tailed).

THSD7A-immunized WT littermates and was slightly positive in some $C3^{-/-}$ mice (Fig. 5g and Supplementary Fig. 16g), which we interpreted as positive staining of trapped circulating complement components in the context of proteinuria and glomerular filtration barrier damage. We next assessed podocyte foot process effacement using the podocyte exact morphology measurement procedure[38]. Slit diaphragm

lengths per area were significantly longer in THSD7A-immunized $C3^{-/-}$ mice than in WT littermate controls, indicating a partial protection from podocyte foot process effacement in $C3^{-/-}$ mice (Fig. 5h, i). Notably, in WT littermates the degree of glomerular complement-activating mIgG subclasses correlated with C5b-9 deposition, C5b-9 deposition correlated with podocyte foot process effacement, and

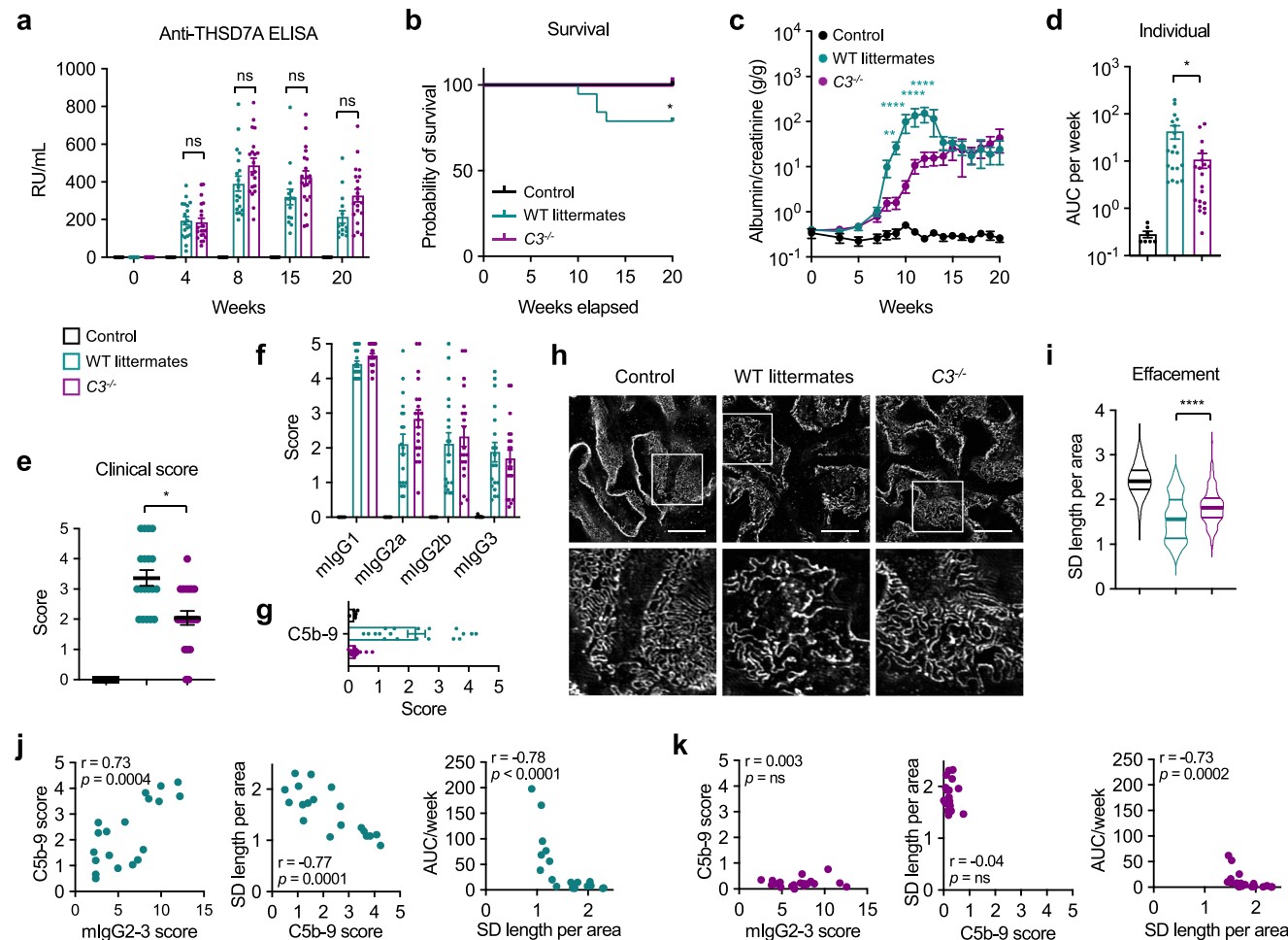

**Fig. 5 | Complement C3-deficient (C3⁻/⁻) mice show an ameliorated course of MN.** **a** Anti-THSD7A antibody titers as measured by ELISA in immunized animals. Controls (n = 7), THSD7A-immunized wild-type (WT) littermates (n = 19) and THSD7A-immunized C3⁻/⁻ mice (n = 20) were experimented in three independent experiments. ns, not significant (mixed-effects analysis with Bonferroni correction for multiple comparisons). **b** Survival analysis. *p = 0.0457 (log-rank Mantel cox test). **c** Albuminuria over time as measured by albumin-to-creatinine ratio. **p = 0.0015, ****p < 0.0001 (mixed-effects analysis with Bonferroni correction for multiple comparisons). **d** Individual albuminuria as measured by the area under the curve (AUC) per week. *p = 0.0384 (Kruskal–Wallis with Dunn's correction for multiple comparisons). **e** Nephrotic syndrome clinical score. *p = 0.0178 (Kruskal–Wallis test with Dunn's correction for multiple comparisons). **f** Quantitative analysis (histological score) of glomerular mIgG subclass deposition. **g** Quantitative analysis (histological score) of glomerular C5b-9 deposition. In panels **b**–**g**, data are presented as mean and SEM and derived from n = 7 controls, n = 19 THSD7A-immunized WT littermates, and n = 20 THSD7A-immunized C3⁻/⁻ mice. **h** Representative images of podocyte foot process effacement as indicated by nephrin staining in structured-illumination super-resolution microscopy. Bars 10 μm. **i** Quantification of podocyte foot process effacement as measured by slit diaphragm (SD) length per area using the podocyte exact morphology measurement procedure (n = 3 controls, n = 19 WT littermates, n = 20 C3⁻/⁻ mice analyzed). Data are presented as mean and SEM. ****p < 0.0001 (Kruskal–Wallis test with Dunn's correction for multiple comparisons). **j, k** Correlation of the cumulative mIgG2a, mIgG2b, and mIgG3 (referred to as mIgG2-3) score with the C5b-9 score (left), correlation of the C5b-9 score with SD length per area (middle), and correlation of SD length per area with individual albuminuria as measured by the AUC per week (right) in THSD7A-immunized WT littermates (**j**) and C3⁻/⁻ mice (**k**). ns, not significant. Spearman's r correlation coefficient (two-tailed).

podocyte foot process effacement correlated with proteinuria (Fig. 5j). By contrast, in C3⁻/⁻ mice only podocyte foot process effacement correlated with individual proteinuria (Fig. 5k). Taken together, these experiments demonstrate a role of C3 in the mediation of podocyte injury and proteinuria, but also indicate that glomerular injury and proteinuria can also develop in the absence of C3 and C5b-9 deposition.

## Morphological subgroup analyses of glomerular alterations in experimental autoimmune MN

Among the WT group immunized during model establishment and the WT littermate group immunized in comparison to C3⁻/⁻ mice (further referred to as WT/WT littermates), but not among C3⁻/⁻ mice, there was a subgroup of animals with severe proteinuria and progressive fluid retention that required an early euthanization before week 20 (Supplementary Fig. 9b and

Fig. 5b). Interestingly, the mice with the most severe disease showed the most prominent glomerular C5b-9 deposition of all examined animals (Fig. 6a) and as expected, stronger podocyte foot process effacement when compared to less diseased WT/WT littermates and C3⁻/⁻ mice (Fig. 6b). Moreover, severe disease was accompanied by loss of podocytes, as determined by counting of DACH-1 positive cells per glomerular area (Fig. 6c).

Recently, the slit diaphragm proteins nephrin and neph1[39], as well as the actin-associated cytoplasmic protein synaptopodin[26], have been suggested to be targets of complement-mediated injury in MN. In our THSD7A-immunized mice, nephrin and neph1 were substantially downregulated compared to controls (Fig. 6d, e). Notably, loss of both proteins was significantly higher in animals with exacerbated disease than in WT/WT littermates with a milder disease and C3⁻/⁻ mice. By contrast, synaptopodin expression was found to be variable, but not significantly

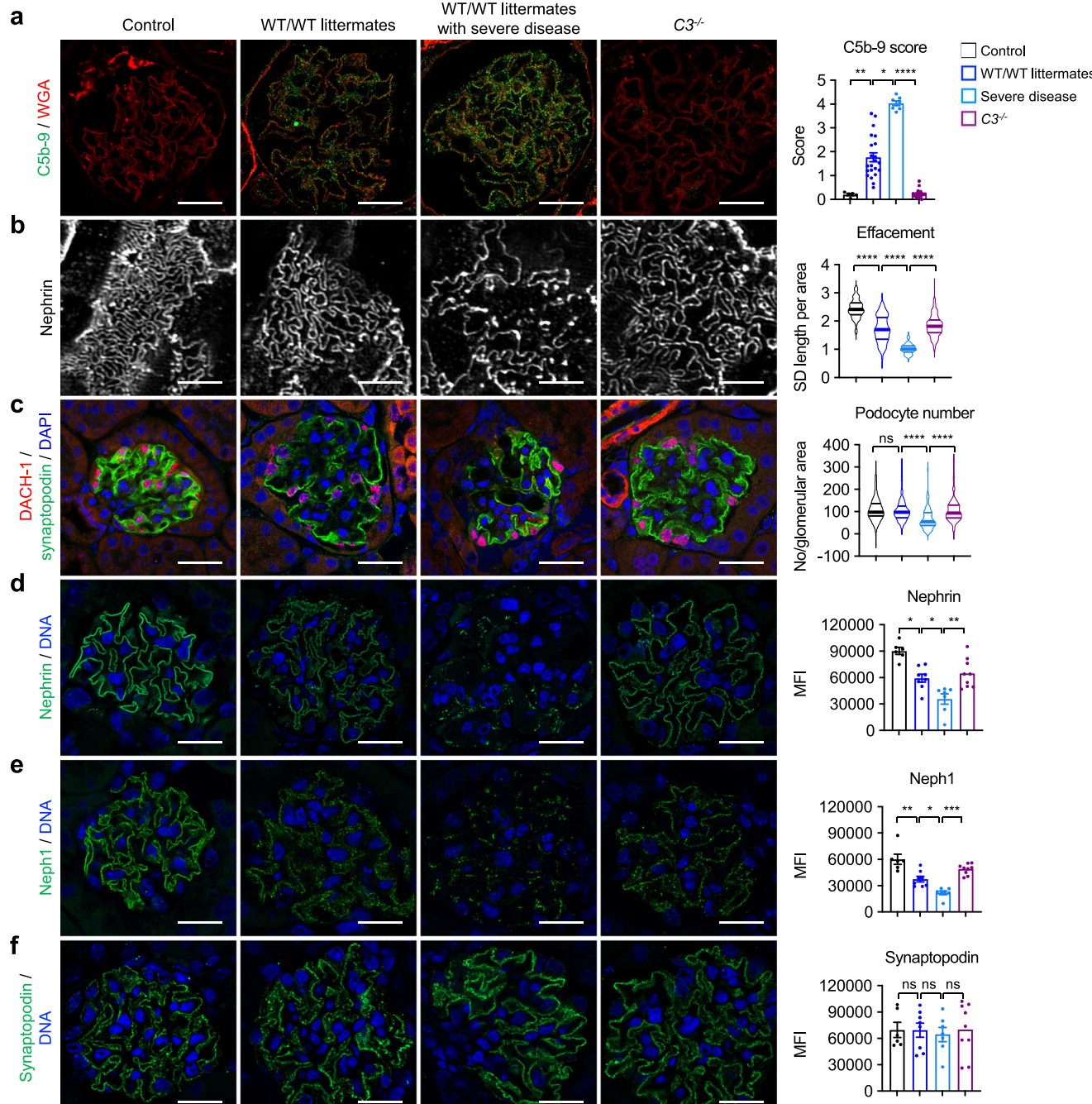

**Fig. 6 | Exacerbation of disease in complement-competent mice is associated with distinct glomerular alterations. a** Representative immunofluorescence stainings and quantification of C5b-9 in control mice ($n = 5$), THSD7A-immunized wild-type (WT) mice, and WT littermates (analyzed together in one group referred to as WT/WT littermates; $n = 22$), THSD7A-immunized WT/WT littermates with severe disease ($n = 7$), and $C3^{-/-}$ mice ($n = 20$). Data are presented as mean and SEM. *p* from left to right, 0.003, 0.0320, <0.0001 (Kruskal–Wallis test with Dunn's correction for multiple comparisons). **b** Representative nephrin stainings in structured-illumination super-resolution microscopy and quantification of podocyte foot process effacement as measured by slit diaphragm (SD) length per area using the podocyte exact morphology measurement procedure. Controls, $n = 3$. WT/WT littermates, $n = 22$. WT/WT littermates with severe disease, $n = 7$, $C3^{-/-}$ mice, $n = 20$. Data are presented as mean and SEM. ****$p < 0.0001$ (Kruskal–Wallis test with Dunn's correction for multiple comparisons). **c** Representative immunofluorescence stainings showing DACH-1–positive podocytes and quantification of podocyte number. Controls, $n = 5$. WT/WT littermates, $n = 21$. WT/WT littermates with severe disease, $n = 7$, $C3^{-/-}$ mice, $n = 20$. Data are presented as mean and SEM. ns, not significant; ****$p < 0.0001$ (Kruskal–Wallis test with Dunn's correction for multiple comparisons). **d** Representative immunofluorescence stainings and quantification (mean fluorescence intensity, MFI) of nephrin. Data are presented as mean and SEM. *p* from left to right, 0.0175, 0.0279, 0.0053 (Kruskal–Wallis test with Dunn's correction for multiple comparisons). **e** Representative immunofluorescence stainings and quantification (MFI) of neph1. Data are presented as mean and SEM. *p* from left to right, 0.0077, 0.0482, 0.0003 (Kruskal–Wallis test with Dunn's correction for multiple comparisons). **f** Representative immunofluorescence stainings and quantification (MFI) of synaptopodin. Data are presented as mean and SEM. ns, not significant (Kruskal–Wallis test with Dunn's correction for multiple comparisons). In **d**–**f** controls, $n = 6$; WT/WT littermates, $n = 8$; WT/WT littermates with severe disease, $n = 7$; $C3^{-/-}$ mice, $n = 9$. Bars in **b** are 10 μm, all other bars are 20 μm.

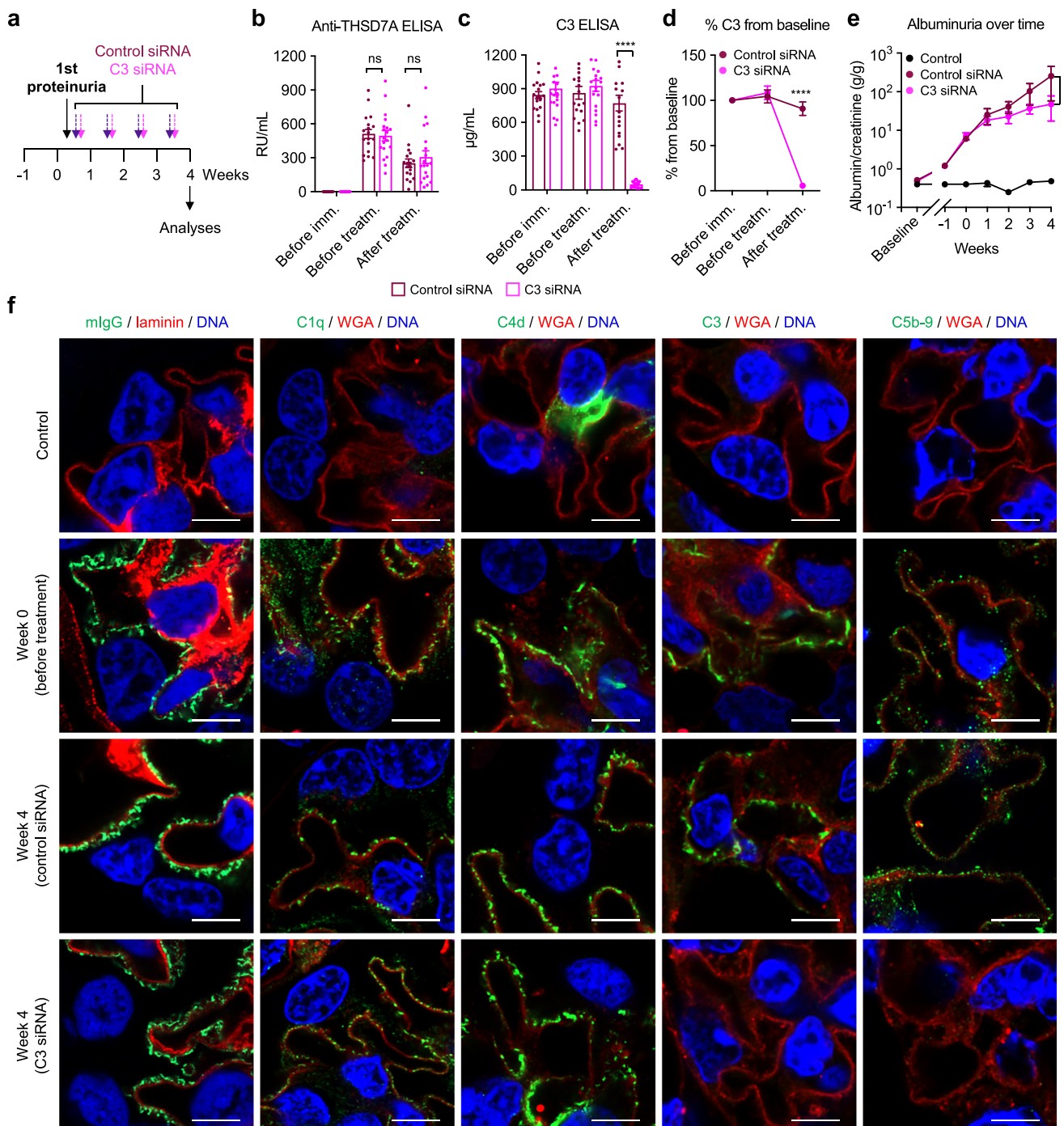

**Fig. 7 | C3-targeted siRNA treatment after onset of proteinuria attenuates MN in mice. a** Experimental scheme. Mice were injected weekly with control siRNA ($n = 18$) or C3 siRNA ($n = 19$) for 4 weeks in three independent experiments, starting when albuminuria exceeded 3 g/g urinary albumin-to-creatinine ratio. **b** Serum anti-THSD7A antibody titers as measured by ELISA. Data are presented as mean and SEM. ns, not significant (two-way ANOVA with Bonferroni correction for multiple comparisons). **c**, **d** Serum C3 levels in µg/mL (**c**) and expressed as percent from baseline (**d**). Data are presented as mean and SEM. ****$p < 0.0001$ (two-way ANOVA with Bonferroni correction for multiple comparisons). **e** Albuminuria over time as measured by albumin-to-creatinine ratio in control mice and THSD7A-immunized mice treated with control siRNA or C3 siRNA. *$p = 0.0127$ (two-way ANOVA with Bonferroni correction for multiple comparisons). In panels **b**–**e**, $n = 18$ control siRNA-treated animals and $n = 19$ C3 siRNA-treated animals were investigated. **f** Representative immunofluorescence stainings for murine IgG (mIgG) and complement components C1q, C4d, C3, and C5b-9 (controls, $n = 3$. Week 0, $n = 1$. Week 4 control siRNA, $n = 6$. Week 4 C3 siRNA, $n = 6$). Bars 4 µm.

decreased in THSD7A-immunized mice (Fig. 6f). In summary, these analyses suggest that extensive complement activation exacerbates podocyte injury through podocyte loss and compromising of the integral slit diaphragm proteins nephrin and neph1[40].

## C3-targeted treatment ameliorates the course of experimental autoimmune MN
RNA interference is a molecular process in which gene expression is silenced through degradation of the target mRNA. For its application as a treatment strategy, small interfering RNA (siRNA) molecules,

which consist of a nucleotide sequence complementary to the target mRNA, are used. Binding of the siRNA to the target mRNA leads to the selective degradation of the mRNA, thus preventing expression of the target protein. Based on the finding that $C3^{-/-}$ mice were partially protected from glomerular injury in experimental autoimmune MN, we hypothesized that a C3-targeted treatment would be beneficial in this model. N-acetyl-galactosamine (GalNAc) conjugated siRNAs targeting murine C3 were utilized to silence C3 in the liver, its primary site of expression (Supplementary Fig. 17a–d). Mice were immunized according to the previously established protocol and assessed weekly for the development of proteinuria. When albuminuria exceeded 3 g/g albumin/creatinine (corresponding to an ~8-fold increase in urinary albumin excretion from baseline), mice received control ($n = 18$) or C3 siRNA ($n = 19$) weekly for a period of 4 weeks (Fig. 7a). Neither group differed in the anti-THSD7A serum titers measured before immunization, before initiation of treatment and after 4 weeks of treatment (Fig. 7b). However, serum C3 levels substantially decreased after 4 weeks of C3 siRNA treatment when compared to control siRNA treatment (Fig. 7c). Compared with baseline C3 levels, the decrease in C3 siRNA was found to be 95% at the end of follow-up in the C3 siRNA-treated group (Fig. 7d). Serum C3 levels also slightly decreased in the control group, likely due to urinary loss of C3. While proteinuria was similar in both groups at the time of treatment start, control siRNA-treated animals developed higher urinary albumin-to-creatinine ratios over time, which reached statistical significance after 4 weeks of treatment (Fig. 7e). Serum albumin levels were higher, and the combined nephrotic syndrome clinical score was lower in C3 siRNA-treated mice than control siRNA-treated mice (Supplementary Fig. 17e, f). One mouse in each treatment group had severe MN with ascites at the end of follow-up, suggesting that C3-targeted treatment did not fully halt disease progression. All THSD7A-immunized mice, but not control mice, had prominent granular, subepithelial deposits of mIgG (Fig. 7f). We found substantial deposition of C1q, C4d, C3, and C5b-9 in one THSD7A-immunized mouse that was analyzed when albuminuria exceeded 3 g/g albumin/creatinine (week 0). Notably, while there was still prominent complement deposition in the control siRNA group after 4 weeks of treatment, C3 and C5b-9 were virtually undetectable in the mice that had received C3 siRNA (Fig. 7f). In summary, our data demonstrate that a 4-week C3-targeted treatment starting after the onset of substantial proteinuria leads to a complete removal of glomerular C3 and the downstream C5b-9 complex and an attenuation of disease in experimental autoimmune MN.

## Discussion

In this study, we systematically analyzed the complement activation pathways in patients with PLA2R1- and THSD7A-associated MN and investigated the pathogenic significance of such complement activation in a novel antigen-specific autoimmune mouse model of MN. Based on the successful detection of key complement components in immunofluorescence analyses, we applied proximity ligation assays to uncover the complement pathways that are activated in patients with MN. We found activity of all three complement initiation pathways, i.e., alternative, classical, and lectin, in the investigated cohort, indicating that complement activation is not restricted to one particular pathway in this disease. Notably, the classical/lectin C3 convertase could be identified in all examined patients with concomitant positivity of the IgGC1q complex in the vast majority of cases, while alternative and lectin pathway components were detected in fewer cases, suggesting predominant complement activation via the classical pathway. This contrasts with several studies in which C1q deposition was a rather infrequent event[3,12–14]. These studies applied direct or indirect immunofluorescence on frozen biopsy samples, which is usually conducted without tissue preprocessing. As we found much higher rates of C1q positivity in our cohort of patients with PLA2R1- and THSD7A-associated MN when investigating paraffin-embedded tissue

involving antigen retrieval, we hypothesized that C1q is concealed in immune deposits in MN[30], but can be unmasked. Indeed, when we performed antigen retrieval using a combination of trypsin and methanol, C1q could be specifically detected in frozen biopsy samples from patients of PLA2R1-associated MN. This finding is in accordance with a recent study, which found C1q to be negative in conventional immunofluorescence but positive in mass spectrometric analysis[41], further supporting the concept of masked C1q deposits in MN.

It appears paradoxical that the dominant IgG subclass in PLA2R1- and THSD7A-associated primary MN is IgG4, the IgG subclass that is considered unable to activate complement by binding of C1q[11]. A recent study has shown that IgG4 with distinct glycosylation patterns can directly activate the complement system through the lectin pathway in vitro[26], providing a potential mechanism of complement activation in IgG4-predominant MN cases. This is in accordance with our finding that glomerular IgG4 deposition closely correlated with IgGMBL signals, suggesting glomerular lectin pathway activation by bound autoantibodies. However, in regard of the observations that (1) MBL was only positive in a subset of cases and with relatively few signals in our study, (2) that not all cases of (primary) MN are positive for IgG4[26,42], and (3) that MN can occur in genetically MBL-deficient patients[27], IgG/MBL interaction appears unlikely to explain the full degree of glomerular complement activity in MN. Our findings that at least one of the C1q-binding IgG subclasses 1–3 was present in 100% of the investigated MN biopsies, that the degree of glomerular IgG1-3 deposition strongly correlated with IgGC1q signals, and that the IgGC1q signals strongly correlated with classical/lectin C3 convertase activity support a central role of classical pathway activation by C1q-binding autoantibodies in patients with PLA2R1- and THSD7A-associated MN. The additional assembly of the alternative C3 convertase in some of the investigated cases can be explained by the amplification loop as a secondary effect and does not conflict a primary, predominant activation via the classical pathway.

The question whether complement activation contributes to podocyte damage in MN has long been an issue of scientific interest. Investigations on molecular pathomechanisms in MN have primarily been conducted in the rat models of passive and active Heymann nephritis (HN). In HN, binding of heterologous (passive HN) or autologous (active HN) antibodies to their podocyte target antigen(s) leads to complement activation with formation of C5b-9, which is inserted into podocyte membranes[43,44]. Two studies found an essential role of complement activation in the mediation of glomerular injury in HN, as indicated by attenuated proteinuria in the setting of C3 and C6 depletion, respectively[45,46]. Moreover, one recent study found a beneficial effect of a small-molecule inhibitor of CFB in passive HN[47]. In contrast, other studies found no impact on proteinuria when complement was removed, challenging this pathophysiological concept[48,49]. More recently, two antigen-specific models of MN were established using passive transfer of rabbit antibodies against THSD7A and PLA2R1[35,50]. Interestingly, while there were some glomerular C3 deposits in the PLA2R1-associated model, C3 was barely detectable in the THSD7A-associated model despite nephrotic-range proteinuria, suggesting a role of complement-independent pathogenic mechanisms in MN. Taken together, the studies in HN and the PLA2R1- and THSD7A-associated passive models do not allow definitive conclusions about the pathogenicity of complement in MN. Importantly, the previously studied experimental models have multiple limitations. First, the classic protocols for passive and active HN involve antibodies against megalin[51], a protein that is not expressed in human podocytes and does not play a role in patients with MN, limiting the specificity of the findings. Second, the heterologous IgG in passive models may have limited effector functions in the new host (e.g., a limited ability to bind host C1q). Third, the heterologous IgG induces an immune reaction with formation of host antibodies against the foreign IgG, which bind to the deposited IgG and exacerbate disease. Fourth, the "antibody

storm" that is caused by transfer of large amounts of heterologous IgG induces a rapid onset of disease, which does not reflect the course of MN in patients, where disease is evolving slowly, likely over the course of weeks to years[52]. These issues were overcome with the model presented in this study, in which immunization of THSD7A-expressing WT mice with immunologically relevant regions of THSD7A led to the generation of anti-THSD7A autoantibodies, which in turn induced all clinical and histological features of MN.

The consequence of the immunization protocol used in this mouse model was a dominant murine IgG1 response. Murine IgG1 does not activate complement and can, in this regard, be considered the murine equivalent of non-complement activating human IgG4, which is the dominant IgG subclass in patients with PLA2R1- and THSD7A-associated MN[4,5,18]. Concurrently, mice also generated complement-activating IgG subclasses, which, like the complement-activating IgG subclasses in MN biopsies, were found to be deposited along the glomerular filtration barrier. The degree of complement-activating IgG deposition correlated with deposition of the membrane attack complex, which in turn correlated with the degree of proteinuria, suggesting a pathogenic role of complement in this model. The antigen specificity and autoimmune character in combination with the similarities between the autoantibody-induced alterations in THSD7A-immunized mice and our findings in MN patients can be considered major strengths of this new model of experimental autoimmune MN. Moreover, the model relies on a straightforward immunization protocol in wild-type BALB/c mice, and we expect it to be easily reproducible and applicable in scientific laboratories. As the model involves the full sequence of immunological and pathophysiological events including antigen presentation, a cellular immune response with activation of B cells and production of autoantibodies, autoantibody binding to the target antigen on podocytes, and induction of local damage with subsequent loss of plasma proteins to the urine, it can be applied to investigate a variety of essential pathophysiological questions in the field of MN. Additionally, this model is suitable for proof-of-concept therapeutic studies, e.g., using treatments targeting autoantibodies, B cells, plasma cells or immune regulation.

Based on our findings that complement activation was multifaceted in both patients with MN and experimental autoimmune MN, we made use of mice deficient in the central complement component C3 ($C3^{-/-}$ mice) to investigate whether complement activation contributes to glomerular damage and proteinuria in MN. Indeed, despite comparable anti-THSD7A titers, $C3^{-/-}$ mice, which showed an interruption of the complement cascade at the level of C3, developed lower overall proteinuria, less hypoalbuminemia, less hyperlipidemia, and less pronounced podocyte foot process effacement than WT littermate controls. Strikingly, a subgroup of mice in the WT littermate group, but not the $C3^{-/-}$ group, developed severe disease with exacerbation of proteinuria and progressive fluid retention, which required early euthanization of these animals. These animals had particularly strong C5b-9 deposition, which was accompanied by extensive podocyte foot process effacement, a decrease in podocyte number, and loss of the integral podocyte slit diaphragm molecules nephrin and neph1. Nephrin and neph1 are critical to maintain integrity of the slit diaphragm and thus to prevent loss of plasma proteins to the urine[53,54].

Treatment of THSD7A-immunized mice after the onset of proteinuria led to an attenuated course of disease. Remarkably, 4 weeks of treatment sufficed to remove glomerular C3 and C5b-9 deposits—which we consider as the essential mechanistic basis for the observed protective effect. Taken together, our experiments in $C3^{-/-}$ mice and in WT mice treated with C3 siRNA demonstrate that activated complement contributes to MN pathogenesis and urinary protein loss, likely by compromising the podocyte slit diaphragm through particularly severe disturbance of nephrin and neph1.

To date, several potent complement inhibitors are in clinical use or under investigation, also for the treatment of MN[55]. Our study further supports the application of complement-targeted treatment in MN, especially for severely affected patients, e.g., with uncontrolled nephrotic syndrome or declining of kidney function. In such cases, targeting the complement system may be beneficial to interrupt an acute, destructive glomerular process. In our opinion, it could be reasonable to apply such a treatment as an add-on strategy together with a therapy targeting the antibody-producing B cells, e.g., rituximab. Such an approach could rapidly reduce complement-mediated glomerular damage and thus lower disease burden until an antibody-lowering therapy takes full effect. However, interference with the complement system at the level of C3 may reduce opsonization with C3b and complement-mediated phagocytosis, which could lead to an increased rate of infections. It deserves special notice that substantial proteinuria also occurred in the setting of a defective complement system, i.e., in $C3^{-/-}$ mice and in mice treated with C3 siRNA. This hints at additional complement-independent mechanisms in the mediation of glomerular damage and proteinuria in MN, which are likely to play an important role. Such complement-independent mechanisms may involve interference of autoantibodies with the biological functions of the target antigen, e.g., enzymatic activity, as it has been demonstrated for NEP-associated MN[56], interaction with other molecules such as focal adhesion proteins, or integral cellular signaling events. We believe that the revelation of such complement-independent pathways will be crucial for an even more profound understanding of the underlying pathomechanisms and for the identification of novel therapeutic targets. Such injury pathways will need to be uncovered in future studies, which may utilize the model described in this study.

Our study has some limitations. First, our functional analyses of the complement system in MN focus on THSD7A as the target antigen and spare PLA2R1. This is due to the circumstance that PLA2R1 is not endogenously expressed by rodent podocytes and thus, experimental models involving PLA2R1 rely on transgenic overexpression of the antigen. However, as we found no substantial differences in complement activation and the deposition of complement-activating IgG between patients with PLA2R1- and THSD7A-associated MN, we speculate that complement plays a similar role in both MN subgroups. Second, there are many distinctive features of the human immune system that cannot be fully reflected in model organisms, such as the mice used in this study. Therefore, uncertainties, for example regarding the role of the different human and murine IgG subclasses and their ability to activate complement, remain. Third, we did not dissect the contribution of each single complement pathway as well as the anaphylatoxins C3a and C5a, whose receptor blockade could also represent promising therapeutic strategies[57], to MN pathogenesis, but rather focused on C3, the central complement component. In light of our data indicating predominant complement activation via the classical pathway, but in some cases also involving the lectin and alternative pathways, we believe that this represents a promising treatment strategy, but future investigations should look at the pathogenic role of each complement pathway specifically.

## Methods
### Patient samples
The study of diagnostic human biopsies was conducted in accordance with federal state and institutional guidelines and approved by the local ethics committee of the Chamber of Physicians in Hamburg (registration number PV5541). Biopsy samples derived from the archive of the Institute of Pathology, University Medical Center Hamburg-Eppendorf, Hamburg, Germany, and were collected between 2017 and 2020. Samples were included based on the histological diagnosis of MN, positive immunohistochemical antigen status for PLA2R1 or THSD7A[5,58], and availability of sufficient biopsy material to perform the analyses for this study. No specific exclusion criteria applied. MN patients had a median age of 63 years and 77% were male (Supplementary Table 1). Samples with a histological diagnosis of

MCD, primary FSGS, diabetic nephropathy, and amyloidosis served as controls. As additional controls, biopsy samples from renal allografts at the time of transplantation were investigated.

## Immunofluorescence

Paraffin sections (3–4 μm) of formalin-fixed paraffin-embedded human or mouse kidneys were deparaffinized and rehydrated to water. Antigen retrieval was obtained by boiling in Dako Target Retrieval pH 9 (Dako, Carpinteria) for 30 min in a steamer at constant 98 °C, or by digestion with protease XXIV (5 μg/mL; Sigma-Aldrich) for 15 min at 37 °C. Unspecific binding was blocked with 5% horse serum (Vector Laboratories) with 0.05% Triton X-100 (Sigma-Aldrich) in PBS for 30 min at room temperature before incubation at 4 °C overnight with primary antibodies in blocking buffer. Staining was visualized with affinity purified, fluorochrome-conjugated secondary antibodies (Jackson ImmunoResearch Laboratories or Invitrogen; 1:200) for 30 min at RT in blocking buffer. Nuclei were counterstained with DRAQ5 (1:1000; Cell Signaling 40482), DAPI (1:400; Sigma-Aldrich), or Hoechst33342 (1:1000; Sigma-Aldrich). Sections were mounted with Fluoromount-G (Invitrogen). Optical images were obtained using the inverted laser confocal microscope LSM800 from Zeiss. Representative images were obtained using the Airyscan with 4-line averages and stored in 1024 × 1024 or 2048 × 2048 pixel frames with 16-bit color depth and the ZEN2.6 (blue edition) software. Enlargements are re-imaged zooms of the overview images (not enlargements of the overview image).

For immunolocalization in human biopsy samples, the following antibodies were used: collagen IV (goat, 1:400, SouthernBiotech 1340-01), human IgG (Cy2 or Cy3 donkey anti-human IgG, 1:200; Jackson ImmunoResearch Laboratories, 709-225-149 and 709-165-149, respectively), C2 (mouse, 1:10; Santa Cruz, sc373809), C4b (rabbit, 1:500; Abcam ab181241), CFB (rabbit, 1:20; Proteintech 10170-1-AP), C3b (mouse, 1:1000; Abcam ab11871), C1q (rabbit, 1:1500; Dako A0136), C1q (goat serum, 1:400; Complement Technology A200), MBL (rabbit, 1:800; Abcam ab190834). C1q stainings using the rabbit anti-C1q antibody were controlled by replacing it with an isotype control in the identical concentration (i.e., 4.7 μg/mL, corresponding to the 1:1500 dilution), followed by incubation with the fluorochrome-labeled secondary antibody. For the localization of human IgG subclasses, monoclonal antibodies against IgG1 (mouse, 1:100, SouthernBiotech, 9052-01, clone no. 4E3), IgG2 (mouse, 1:40,000, SouthernBiotech, 9080-01, clone no. HP6014), IgG3 (mouse, 1:1000, SouthernBiotech, 9210-01, clone no. HP6050), and IgG4 (mouse, 1:4000, SouthernBiotech, 9200-09, clone no. HP6025) were used.

For immunofluorescence analyses on frozen biopsy samples, 6 μm cryo-sections were air-dried. For C1q-unmasking, an adapted immunofluorescence protocol was developed. Briefly, antigen retrieval was performed on air-dried sections using −20 °C MeOH for 5 min. After MeOH evaporation, sections were hydrated using PBS, followed by a short digestion with trypsin (62.5 μg/mL, Sigma) for 10 min at 37 °C. Following washes with PBS, unspecific binding was blocked for 30 min at RT using 0.05% TX-100 diluted in 5% normal horse serum in PBS. Primary antibodies C1q (rabbit, 1:1500; Dako A0136), C1q (goat serum, 1:100; Complement Technology A200), laminin (rabbit, 1:800; Sigma-Aldrich L9393) or collagen IV (1:800, SouthernBiotech 1340-01) were diluted in blocking buffer and incubated overnight at 4 °C. After washes in PBS, primary antibody binding was visualized using appropriate affinity-purified and fluorochrome-labeled donkey secondary antibodies (all Jackson Immunoresearch Laboratories), nuclei were counterstained with Hoechst (1:1000, Molecular Probes). C1q stainings using the rabbit anti-C1q antibody were controlled by replacing it with an isotype control in the identical concentration (i.e., 4.7 μg/mL, corresponding to the 1:1500 dilution), followed by incubation with the fluorochrome-labeled secondary antibody.

For immunolocalization in mice, the following antibodies were used: nephrin (guinea pig, 1:200; Acris Antibodies BP5030), neph1 (guinea pig, 1:200; kindly provided by Florian Grahammer), laminin (rabbit, 1:1000; Sigma-Aldrich L9393), WGA-rhodamin (1:400; Vector-Laboratories), THSD7A (goat, 1:200; Santa Cruz Biotechnology sc163455 or rabbit, 1:200; Sigma-Aldrich HPA000923), C1q (goat serum, 1:400; Complement Technology A200), C3 (FITC goat anti-C3, 1:100; Cappel 55500), C4d (rabbit, 1:50; Hycultec HP8033), C5b-9 (rabbit, 1:200; Abcam ab55811), CFB (goat, 1:50; Complement Technology A235), CFH (goat, 1:100; Complement Technology A237), MBL (rabbit, 1:800; Abcam ab190834), murine IgG (Cy2 donkey anti-mouse IgG H + L, 1:200; Jackson ImmunoResearch Laboratories), murine IgG subclasses (IgG1, IgG2a, IgG2b, IgG3) (goat, 1:5000; Rockland 610-101-040, 610-101-041, 610-101-042, 610-101-043, respectively), synaptopodin (guinea pig, 1:200; Synaptic Systems 163004 or rabbit, 1:400; Santa Cruz 50459), DACH-1 (rabbit, 1:100; Sigma-Aldrich HPA012672), 8-oxoguanine (goat, 1:600; Abcam ab10802).

The following fluorochrome-conjugated secondary antibodies were used (all 1:200, Jackson ImmunoResearch Laboratories): anti-goat IgG AF488 (705-545-147), anti-goat IgG Cy3 (705-165-147), anti-guinea pig IgG Cy5 (706-175-148), anti-mouse IgG Cy2 (715-225-150), anti-mouse IgG Cy3 (715-165-150), anti-rabbit IgG AF488 (711-545-152), anti-rabbit IgG Cy3 (711-165-152).

For quantification of glomerular deposition of murine IgG subclasses and C5b-9, a histological score from 0 (no signal) to 5 (very strong signal) was established using exemplary images from a pilot set of immunofluorescence stainings that were taken during establishment of the staining protocol. Subsequently, mIgG subclasses and C5b-9 were stained in all samples, a minimum of five pictures were taken from each section, and images were scored according to the established score. Stainings, pictures, and scoring were performed in a blinded manner by three different investigators. Nephrin, neph1, and synaptopodin deposition was quantified using ImageJ (64-bit) by circling individual glomeruli (7 per mouse) and then analyzing mean fluorescence intensity.

## Immunohistochemistry

For the detection of human IgG subclasses, paraffin sections (1 μm) were deparaffinized, rehydrated, pretreated with proteinase type 24 (Sigma-Aldrich) for 30 min at 40 °C and blocked with normal horse serum (Vector Laboratories) for 10 min at 37 °C. Monoclonal antibodies against IgG1 (mouse, 1:100, SouthernBiotech), IgG2 (mouse, 1:40,000, SouthernBiotech), IgG3 (mouse, 1:1000, SouthernBiotech), and IgG4 (mouse, 1:4000, SouthernBiotech) were applied overnight at 4 °C. Bound antibodies were detected using the ZytoChem Plus AP Polymer System (Zytomed), visualized with new fuchsin for 30 min at room temperature and counterstained with Mayer's Hematoxylin. For quantification of glomerular IgG subclass deposition, a histological score from 0 (no signal) to 5 (very strong signal) was established using exemplary images from a pilot set of immunohistochemical IgG subclass stainings. IgG subclasses 1 to 4 were then stained in biopsies from patients with PLA2R1- and patients with THSD7A-associated MN and pictures were taken from all glomeruli that were present in the respective biopsy. All images were then scored according to the established score in a blinded manner. If the mean score was >0.5, the sample was defined to be positive.

For the stainings of immune cells in mice, paraffin sections (2.5 μm) of formalin-fixed paraffin mouse kidneys were deparaffinized and rehydrated to water. Antigen retrieval was obtained for the CD3 staining by incubating in Dako Target Retrieval pH 9 (Dako, Carpinteria) for 15 min in a steamer at constant 98 °C and another 25 min at RT, and for F4/80 staining by digestion with trypsin (Sigma T7168) for 10 min at 37 °C. Unspecific binding was blocked with ZytoChem Plus AP Polymer System Blocking Buffer (Zytomed) for 5 min at room

temperature. Polyclonal antibodies against CD3 (rat, 1:1000, Dako A0453) and F4/80 (rat, 1:1000, BMA T-2006) diluted in Zytomed buffer (Zytomed) were applied overnight at 4 °C, followed by the incubation with an anti-rat IgG antibody (rabbit, 1:200, Vector BA-4001) for 30 min at room temperature. Bound antibodies were detected using the ZytoChem Plus AP Polymer System (Zytomed), visualized with new fuchsin for 10–15 min at room temperature and counterstained with Böhmer's Hematoxylin.

### Periodic acid-Schiff (PAS) stain
Paraffin sections (1.5 μm) of experimental mouse kidneys were deparaffinized and rehydrated to water. To oxidize diols to aldehydes, sections were incubated in 1% periodic acid for 15 min. Afterward, aldehydes reacted for 40 min at room temperature with the Schiff reagent (Sigma) to obtain a purple-magenta color. Counterstain of nuclei was performed with hemalaun and following dehydration the sections were mounted with Eukitt (O. Kindler GmbH).

### Electron microscopy
Electron microscopic analyses were performed on kidneys that were fixed in 4% buffered paraformaldehyde. Tissue was post-fixed with 1% osmium in 0.1 M sodium-cacodylat buffer, stained with 1% uranyl acetate and embedded in epoxy-resin (Serva). Ultra-thin sections were cut (Ultramicrotome, Reichert-Jung) and contrasted with uranyl acetate in methanol followed by lead citrate. Micrographs were generated with a transmission-electron microscope (JEM 1010, JEOL).

### Proximity ligation assays
To perform these assays, we made use of diagnostic biopsy samples from a cohort of patients with PLA2R1- and THSD7A-associated MN. Time point zero biopsy samples from renal transplant recipients were used as controls. All MN patients were without previous immunosuppressive therapy at the time of renal biopsy.

Paraffin sections (1–2 μm) were deparaffinized, rehydrated, and pretreated with proteinase (Sigma-Aldrich) for 30 min at 40 °C. Endogenous peroxidases were quenched with $H_2O_2$ (Duolink® In Situ Detection Reagents Brightfield) and samples were blocked using normal horse serum (Vector Laboratories) for 10 min at 37 °C. Antibodies against the classical C3/C5 convertase components C2 (mouse, 1:10; Santa Cruz sc373809) and C4b (rabbit, 1:500; Abcam ab181241) as well as against the alternative C3/C5 convertase C3b (mouse, 1:1000; Abcam ab11871) and CFB (rabbit, 1:20; Proteintech 10170-1-AP) were each subsequently applied overnight at 4 °C. Antibodies against IgG (mouse, 1:7500; Jackson ImmunoResearch Laboratories 209-005-088) and C1q (rabbit, 1:1500; Dako A0136) were applied for 30 min at 37 °C. For the IgGMBL proximity ligation assay, anti-IgG (rabbit, 1:150,000; Jackson ImmunoResearch 309-005-003) was applied for 30 min at 37 °C, followed by anti-MBL (mouse, 1:20; ThermoFisher MA1-40145) overnight at 4 °C. Secondary anti-mouse (Duolink® In Situ PLA® Probe Anti-Mouse MINUS) and anti-rabbit (Duolink® In Situ PLA® Probe Anti-Rabbit PLUS) antibodies were applied according to protocol, followed by the detection brightfield kit (Duolink® In Situ Detection Reagents Brightfield) for visualization. Nuclear staining was performed using Mayer's Hematoxylin. Whole slide images (Zeiss Axio Scan.Z1) were analyzed using explainable AI (xAI) image analysis software (HSA KIT, HS Analysis GmbH, Karlsruhe, Germany)[31], using the combination of Machine Learning as the Maximally Stable Extremal Regions (MSER) algorithm and Deep Learning as Convolutional Neural Networks (CNNs). The results of module PLA Analysis in HSA KIT were calculated with MSER as signal densities, defined as numbers of signals per glomerular area (in mm²). The human machine-collaboration integrated in HSA KIT gives the feedback loops to ensure the validity of the proximity ligation assay quantification in visual and statistical manner. Values above the mean of the controls plus three times the standard deviation (s.d.) of the controls (mean + 3x s.d. of controls) were defined to be positive.

### Animal experiments
Animal experiments were performed according to national and institutional animal care and ethical guidelines and were approved by the Veterinarian Agency of Hamburg and the local animal care committee (registration numbers 114/18 and 002/19). Mice were housed in a specific-pathogen-free facility at temperatures of 21–24 °C with 40–70% humidity on a 12 h light/12 h dark cycle and provided with food and water ad libitum.

Wild-type male BALB/c[32,35] and C57BL/6[35] mice (12 weeks old) were bred in the animal facility of the University Medical Center Hamburg-Eppendorf. Animals had free access to water and standard animal chow. All animals, i.e., experimental and control mice, were co-housed. Mice were euthanized by perfusion.

To obtain C3 knockout ($C3^{-/-}$) mice on the BALB/c background, a breeding pair of B6;129S4-C3$^{tmCrr}$/J (stock no. 003641) was purchased from The Jackson Laboratory (Farmington, CT, USA). Their littermates were backcrossed ten times to wild-type WT BALB/c, resulting in a C.Cg.C3$^{tm1Crr}$ mouse line. To obtain animals for the experiments, mice with a heterozygous C3 status were bred to obtain homozygous $C3^{-/-}$ mice as well as WT littermates.

Mice lacking THSD7A ($Thsd7a^{-/-}$ mice) were generated using the knockout-first strategy[59]. Thsd7a knockout first mice ($Thsd7a^{tm1a}$ mice) were obtained from the Mutant Mouse Resource & Research Centers, MMRRC, stock no. 063558-UCD. Since $Thsd7a^{tm1a}$ mice showed a residual renal THSD7A expression of approximately 10%, two further breeding steps were necessary to obtain a $Thsd7a^{-/-}$ mouse line. First, $Thsd7a^{tm1a}$ mice were bred with Flip-deleter mice, which constitutively express the Flip-recombinase, leading to $Thsd7a^{tm1c}$ mice with an almost reconstituted THSD7A expression. Second, $Thsd7a^{tm1c}$ were bred with Cre-deleter mice, constitutively expressing the Cre recombinase, leading to $Thsd7a^{tm1d}$ mice with an efficient removal of exon 4 of THSD7A. In contrast to $Thsd7a^{tm1a}$ mice, $Thsd7a^{tm1d}$ mice, or $Thsd7a^{-/-}$ mice, had no detectable renal THSD7A expression. $Thsd7a^{-/-}$ mice were crossed to the BALB/c background for ten generations.

For immunizations, a mixture of four murine THSD7A fragments (d1_d2, d10_d11, d14_d15, d16_d17) was used. Mice received a total of four immunizations: 20 μg of protein per fragment were mixed with an equal volume of complete Freund's adjuvant and injected subcutaneously, followed by three boost immunizations of 20 μg per fragment in an equal volume of incomplete Freund's adjuvant after 3, 5, and 7 weeks. Control mice received the equal amounts of adjuvant diluted in PBS. A total of 15 male BALB/c mice (5 with PBS and 10 with THSD7A domains) were experimented in two independent cohorts (cohort A1: 8 mice, cohort A2: 7 mice). A total of 12 C57BL/6 mice (2 with PBS and 10 with THSD7A domains) were experimented in one cohort. As an additional control, mice were immunized with ovalbumin using the same protocol as described above.

A total of 20 male $C3^{-/-}$ mice and 19 WT littermates were immunized with THSD7A fragments and a total of 2 $C3^{-/-}$ mice and 5 WT littermates were immunized with PBS as described above in three independent cohorts (cohort B1: 10 mice, cohort B2: 20 mice, cohort B3: 16 mice).

Serum was taken before the first immunization and after 4, 8, and 15 weeks. Urine was collected using metabolic cages and mice were weighed weekly. Albumin content in both urine and serum was quantified using a commercially available ELISA system (Bethyl) according to the manufacturer's instructions. The urine albumin values were standardized against urinary creatinine values (as determined according to Jaffé) of the same sample. BUN, triglycerides, and cholesterol levels were measured by standard procedures using the cobas system (Roche Diagnostic). Mice were euthanized after the predefined observation period of 20 weeks for the final collection of blood, urine, and organs, or if criteria for an early termination of the experiment were fulfilled. These criteria included, amongst others that did not apply in our experiments, development of visible ascites, respiratory

distress due to fluid accumulation, and a weight gain of >20% in 1 week. A nephrotic syndrome clinical score was calculated for each mouse. This score involved proteinuria of >10 g/g (1 point), proteinuria >100 g/g (1 point), a serum albumin level >20% below control level (1 point), a serum cholesterol or triglyceride level of >50% above control level (1 point), and a weight gain of >20% in comparison to the weight at week 0 (1 point), thus allowing a maximum score of 5 points. Animal experiments were approved by the veterinarian agency of Hamburg and the local animal care committee.

An siRNA was designed to specifically target a region of the mouse C3 gene using the following sequence: 5′-AACAAGAAGAACAAACU-CACA-3′ (passenger strand), 5′-UGUGAGUUUGUUCUUCUUGUUCA-3′ (guide strand). The siRNA is fully modified with 2′-O-Methyl and 2′-Flouro stabilizing modifications and conjugated to a tri-antennary GalNAc on the 3′ end of the sense strand to enable liver targeting (Alnylam Pharmaceuticals, Cambridge, USA)[60]. siRNAs were injected subcutaneously at a dose of 15 mg/kg body weight. In preliminary experiments to investigate the efficacy in our mice, 6 WT BALB/c mice received two injections of C3 siRNA and 6 animals received two injections of an irrelevant control siRNA targeting luciferase at an interval of 7 days and were analyzed 7 days after the second injection. Blood to measure C3 and C5 levels was taken before the first and before the second injection of siRNA. For treatment experiments in THSD7A-immunized mice, mice received a total of four subcutaneous injections of 15 mg/kg of control siRNA ($n = 18$) or C3 siRNA ($n = 19$) at an interval of 7 days, starting when proteinuria exceeded 3 g/g albu-min-to-creatinine, which was defined as week 0. A total of 6 mice were PBS-immunized as baseline controls. Experiments were performed in three independent cohorts (cohort C1: 14 mice, cohort C2: 18 mice, cohort C3: 11 mice). One additional THSD7A-immunized mouse was euthanized at week 0 to assess glomerular complement deposition at the time of treatment initiation. Mice were euthanized for histological and serum analyses after a predefined observation period of 4 weeks from the first siRNA administration.

## Model-based stereology
Model-based stereology for podocyte morphometric analysis has been previously described[61]. An average of 20 glomeruli per mouse with a total of 625 glomeruli were analyzed. Podocytes were defined as glo-merular cells co-expressing synaptopodin as a cytoplasmatic and DACH-1 + DAPI as nuclear markers.

## Podocyte exact morphology measurement procedure (PEMP)
To quantify foot process morphology, analysis of filtration slit density was performed[38,62]. 3-µm paraffin sections mounted on high-precision coverslips coated with Poly-L-lysine were deparaffinized and antigen retrieval was performed by boiling at 98 °C in Tris/EDTA target retrieval buffer, pH 9 (Agilent Dako; #S2367) for 40 min. Unspecific binding was blocked in 5% horse serum with 0.05% Triton X-100 for 30 min at RT. Incubation with anti-nephrin (guinea pig, 1:200; Progen #GP-N2 1:200) in blocking buffer as primary antibody was performed overnight at 4 °C. Binding was visualized by incubation with affinity-purified Cy2-conjugated donkey anti-guinea pig IgG (1:200; Jackson ImmunoResearch Laboratories; #706-225-148) diluted in blocking buffer for 30 min at RT. Stained sections were mounted with ProLong Gold for imaging. 3D-SIM z-stacks of nephrin-stained kidney sections were acquired with a Zeiss Elyra PS.1 SIM microscope using the ZEN software (all Zeiss, Jena, Germany) with five horizontal shifts and five rotations of the illumination pattern with a slice-to-slice distance of 0.3 µm. Reconstruction of 3D-SIM images was performed using the Zen Black software. PEMP analysis was performed using the PEMMP macro for Fiji (2). For this, the capillary area was encircled, and the slit dia-phragm length was determined. Filtration slit density (FSD) values were calculated from the ratio of slit diaphragm length and capillary area. Per animal, 3–5 glomeruli with 2–6 regions of interest were

analyzed by PEMP. Two-tailed t-test was performed for statistical analysis. Differences at a $P < 0.05$ were defined as statistically significant.

## Design and generation of THSD7A variants
For active immunization, we designed fragments of murine THSD7A (mTHSD7A) containing two consecutive thrombospondin repeat domains (TSRs) each. These fragments were orthologous to the regions in human THSD7A that are most frequently recognized by human autoantibodies present in sera from patients with THSD7A-associated MN. Four fragments were designed: d1_d2 (Ala-48 to Gln-192), d10_d11 (Val-696 to Gln-831), d14_d15 (Lys-960 to Asn-1095), and d16_d17 (Gln-1096 to Tyr-1220). All variants were generated by PCR (Supplementary Table 2) and cloned into the eukaryotic expression vector pCSE2.5 (provided by Thomas Schirrmann, Braunschweig, Germany). This vector has been optimized for secretory protein production in suspension cultures of HEK293-6E cells. The cDNA of a full-length, flag-tagged mTHSD7A construct served as the PCR template (Origene). All constructs were designed to be secreted to the cell culture medium and contained a C-terminal 6x his-tag. For serum analysis of mice, an additional construct comprising all mTHSD7A extracellular domains with an N-terminal strep-tag was designed (d1_d21, Supplementary Table 2) and cloned into the eukaryotic expression vector pDSG-IBA104 (IBA-lifescience). Full sequencing validated the accuracy of all constructs.

## Cell culture, cell transfection, recombinant protein expression, and purification
The THSD7A fragments and the flag tagged full-length variant were expressed in human embryonic kidney cells (HEK). For soluble proteins, HEK293-6E cells (kindly provided by Yves Durocher, Ottawa, Canada[63]) were cultured in cell culture bottles with 30 ml of serum-free medium (Freestyle 293, Gibco). For transfection of HEK293-6E cells, a polyethylenimine (PEI)-based method (Polyscience Inc.) was used. For each approach, 126 µL (40 µg) of PEI was mixed with 124 µL water and 250 µL NaCl (300 mM) was added. 10 µg Plasmid DNA in 250 µL water was mixed with the same amount of NaCl in a separate tube. Subsequently, solutions were slowly mixed, vortexed, and incubated for 30 min. The solution was then added to the cells. After 24 h, the cells were fed with 250 µL feeding medium (Freestyle 293 medium supplemented with 20% tryptone). Five days later, cells were collected using centrifugation at $300 \times g$ and the supernatant was centrifuged again at $14,000 \times g$ for 10 min. For the full-length mouse THSD7A construct, HEK293T cells (derived from the DSMZ-German Collection of Microorganisms and Cell Cultures GmbH, product number ACC 635) were transfected, using a calcium phosphate-based method. Briefly, 10 µg of plasmid DNA was mixed with 36 µL of 2 M CaCl₂ and diluted with sterile water up to a volume of 300 µL. This solution was gently mixed with an equal volume of 2x HEPES buffered saline (HBS, 275 mM NaCl, 55 mM HEPES, pH 7.0) and incubated 30 min at room temperature. The resulting solution was added drop wise to the cells. Medium was changed 24 h after transfection. After 48 h, cells were scraped, centrifuged at $240 \times g$ for 5 min, washed with PBS, and centrifuged again at $240 \times g$ for 5 min. Cells were then lysed in 50 mM Tris pH 7.4, 1 mM EDTA, 150 mM NaCl, 1% Triton after addition of a protease inhibitor cocktail (Roche Diagnostics), sonicated, and centrifuged at $14,000 \times g$ and 4 °C for 1 h. The supernatant was kept, and protein concentration was determined using the Pierce BCA protein assay kit according to the manufacturer's instructions (Thermo Scientific). His-tagged mTHSD7A fragments were purified under native conditions using a Ni-NTA resin (Thermo Scientific) applying the batch method according to the manufacturer's instructions. After purification, the buffer was exchanged to PBS using ZEBA-spin columns (Thermo Scientific). The d1_d21 construct was purified under native conditions using a strep-tactin resin, following the manufacturer's instructions

(Strep-Tactin XT, IBA-lifescience). All expressions and purifications were validated by Western blot and/or Coomassie staining.

## Antibody elution and western blotting
Three hundred 10-μm sections from frozen renal tissue were resuspended in 1 mL sterile PBS and washed three times in PBS. In a first elution step, the supernatant was discarded and the pellet was resuspended in 100 μL of 25 mM citrate, pH 3.2, and left on ice for 20 min with intermediate shaking. The sample was centrifuged with low speed and the supernatant was added to 150 μL of 1 M Tris, pH 8.5. In a second elution step the pellet was resuspended in 100 μL 25 mM citrate, pH 2.5, and left on ice for 20 min with intermediate shaking. Subsequently, the sample was centrifuged for 5 min at $20.817 \times g$ at 4 °C. The supernatant was added to the first elution and diluted to 2 mL using Superblock Blocking Buffer in TBS (Thermo scientific). The resulting sample was used as the primary antibody in immunoblot analyses on mouse glomerular extracts and recombinant full-length mTHSD7A. Mouse glomerular extracts were prepared by means of perfusion of the renal arteries using Dynabeads (Dynal). Protein samples were mixed with 5x Laemmli buffer (1.5 M Tris-HCl, pH 6.8, 50% glycerol, 10% SDS, 1% bromophenol blue), heated at 95° for 10 min, and separated in precast gradient gels (4–15% TGX, BioRad) using SDS-PAGE. Subsequently, the samples were transferred to methanol-soaked PVDF membranes (EMD Millipore) under semi-dry conditions. Membranes were blocked with 3% dry milk with TBS-Tween 0.05% (TBS-T) at RT for 2 h followed by incubation with the antibody elution at 4 °C overnight. Membranes were then washed 3x in TBS-T, incubated with HRP-conjugated goat anti-mouse IgG in 3% dry milk with TBS-T (1:20,000; Jackson ImmunoResearch Laboratories) for 2 h at RT, and washed again. To visualize binding of the eluted antibodies, membranes were incubated in chemiluminescent substrate (SuperSignal West Pico; Thermo Scientific) for 5 min followed by incremental luminescence detection with an Amersham Imager 600.

## ELISA
For the THSD7A ELISA, microplates (Sarstedt, high bind) were coated with 100 ng of the extracellular region of murine THSD7A (d1_d21, Supplementary Table 2) per well in carbonate-bicarbonate buffer overnight at 4 °C, washed three times with washing buffer (Tris-buffered saline with Tween 20, pH 8), and blocked with BSA in Tris-buffered saline, pH 8, for 30 min. Mouse sera were diluted 1:100 or 1:200 in Tris-buffered saline with BSA, pH 8, and incubated for 2 h followed by three washing steps with washing buffer. Bound antibodies were detected by incubation with HRP-conjugated goat anti-mouse IgG antibody (1:5000; Jackson ImmunoResearch Laboratories), diluted in Tris-buffered saline with BSA, pH 8, for 1 h and subsequently washed for 4 times. Then, the TMB substrate (Aviva systems biology) was added for 2 min. The reaction was stopped with 1 M $H_3PO_4$. All incubation steps were carried out at room temperature. The optical density (OD) was read at 450 nm using an automated spectrophotometer (Ultra-microplate reader, Bio-TEK instruments).

A mixture of highly positive sera, derived from THSD7A knockout (*Thsd7a*$^{-/-}$) mice that were immunized with murine THSD7A was used to generate a standard curve consisting of five calibrators (4, 20, 100, 500, and 1500 relative units [RU] per mL). All samples with an absorption value lower than the one for the lowest calibrator (4) were set as zero. The standard dilution for mouse sera was 1:100. Samples with an absorption above the highest calibrator (1500) were re-analyzed at a 1:200 dilution.

To analyze the distribution of the different IgG subclasses among the anti-THSD7A IgG, sera of all available time points were analyzed. The amount of coated protein, all buffers as well as the detection procedure were as described in detail above. Sera were used in a 1:50 dilution. After washing, antibodies against the different subclasses were added (IgG1, IgG2a, IgG2b, IgG3; all goat, 1:5000; Rockland 610-101-040, 610-101-041, 610-101-042, 610-101-043, respectively) and incubated for 1 h, followed by three washing steps. To detect the subclass antibodies, a HRP-conjugated donkey anti-goat IgG antibody (1:5000; Jackson ImmunoResearch Laboratories) was used. The optical density (OD) at 450 nm was analyzed. For detection of anti-ovalbumin antibodies, 100 ng of ovalbumin was coated and mouse sera were tested in a 1:100 dilution as described in detail above. The optical density (OD) at 450 nm was analyzed.

## Analysis of C3 and C5 mRNA and protein levels
Liver C3 and C5 mRNA levels were quantified by RT-qPCR. Briefly, RNA was extracted from liver powders using the RNeasy Mini Kit according to manufacturer's protocol (Qiagen, Inc). Reverse transcription to generate cDNA was performed according to manufacturer's protocol (Life Technologies). qPCR was performed on cDNA using a Roche LightCycler 480 instrument utilizing mouse C3 and C5 Taqman FAM probes Mm01232779_m1 (C3) and mm00439275_m1 (C5) (Life Technologies) and glyceraldehyde-3-phosphate dehydrogenase (GAPDH) Taqman VIC probe (Life Technologies) as control. C3 and C5 levels were normalized to GAPDH and percent C3 and C5 mRNA remaining was calculated relative to the average of control siRNA-treated mice.

Serum C5 levels were quantified as previously described[64]. Serum C3 was quantified using a mouse C3 ELISA kit (ab157711, Abcam) according to the manufacturer's instructions.

## Statistics
Animals were assigned to treatment groups randomly and age-matched between conditions where necessary. If quantifications were performed using histological samples, staining, imaging, and scoring procedures were performed in a blinded manner by three different researchers. No animals were excluded from the analyses. Details on experimental replicates and animal numbers are given above. Statistical analysis was performed using GraphPad Prism (La Jolla, CA). The results are shown as scatter dot plots with a bar indicating the mean ± SEM. Differences between two groups were analyzed using a two-tailed Mann–Whitney test. Differences between three groups were calculated using a Kruskal–Wallis test with a Dunn's multiple comparisons test. Comparisons with multiple variables were analyzed using a two-way ANOVA with Bonferroni's multiple comparisons test. In case of missing values, differences were analyzed using a mixed-effects analysis with Bonferroni's multiple comparison test. Correlation analyses were performed using the nonparametric Spearman correlation.

## Reporting summary
Further information on research design is available in the Nature Portfolio Reporting Summary linked to this article.

## Data availability
All data supporting the findings of this study are included in the main article, the supplementary information, and the source data that are provided with this manuscript, including fluorescence quantifications and uncropped gels/blots. Source data are provided with this paper.

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

## Acknowledgements

This work was funded by the Deutsche Forschungsgemeinschaft as part of the Emmy Noether Programme "Molecular Mechanisms of Membranous Nephropathy" (TO10-13) to N.M.T. This work was additionally funded by the Deutsche Forschungsgemeinschaft as part of the Sonderforschungsbereich 1192 to N.M.T. and T.B.H. (project B2), C.M.S. (project B3), F.K.N. (project B5), T.W. and P.F.Z. (project B6), V.G.P. (project B9), and U.P. (project A2). Further funding was provided by KidNeeds, Iowa City, Iowa, USA, to P.F.Z. and the Else Kröner Fresenius Stiftung to S.F. and C.M.S., the Bundesministerium für Bildung und Forschung (eMed Consortia "Fibromap VGP") and the Novo Nordisk Foundation (Young Investigator Award; NNF21OC0066381) to V.G.P., and by the Deutsche Forschungsgemeinschaft to O.K. (KR1984/4-1).The authors thank Josephine Gebhardt, Institute of Immunology, for excellent technical assistance with the production of recombinant proteins in HEK293-6E cells.

## Author contributions

N.M.T. conceptualized and supervised the study, designed and performed experiments, analyzed and interpreted the data, and wrote the manuscript. T.W. developed proximity ligation assays, analyzed and interpreted the data and co-supervised the study. L.S. performed most of the experiments. G.Z., N.H., C.M.S., S.D., S.W., S.M.S.K., R.L., D.K., S.F., S.Z., O.K., A.B., S.B., Y.W., H.C., and V.G.P. performed experiments. N.M.T., T.W., A.B., H.C., P.F.Z., H.H., U.P., and T.B.H. provided tools and materials. P.F.Z., A.B., F.K.N., U.P., and T.B.H. provided expertise and feedback. N.M.T., T.B.H., T.W., P.F.Z., C.M.S., F.K.N., V.G.P., and U.P. provided funding. All authors reviewed and edited the manuscript.

## Funding

## Competing interests

A.B. is an employee and shareholder of Alnylam Pharmaceuticals. P.F.Z. serves as a consultant for Eleva GMBH, Novartis, Generic Assays, Alexion, Bayer, Vifor Fresenius Medical Care Renal Pharma, and Samsung Bioepis. All other authors declare no competing interests. P.F., S.W., and T.W. applied for a patent "C3/C5 convertase assays" (EP 3771468/US-2022-0276261).
