## [Peer Review File · Nature Communications]

REVIEWER COMMENTS

Reviewer #1 (expertise in nephrology, glomerular disease):

Seifert et al. evaluated complement cascade activation in a model of membranous nephropathy (MN) and found that all 3 pathways (classical, lectin, and alternative) are activated. They also found a close correlation to the degree of glomerular C1q-binding IgG subclasses. Complement-deficient mice were protected from glomerular damage. Targeting of C3 using RNA interference attenuated disease. The authors conclude that complement is primarily activated via classical pathway and that complement components such as C3 represent promising therapeutic targets in MN.

Comments:

1. The statement that C1q is present in human biopsies of patients with MN is puzzling since this reviewer has seen multiple biopsies of patients with MN and never saw C1q staining, not even when pronase has been done, except of course in cases of membranous lupus nephropathy. Similarly, mass spectrometry data does not show C1q to be present (Ref #22). Please provide additional enlarged immunofluorescence biopsy pictures of at least 10 of the human primary MN (not lupus membranous) showing C1q staining (additional patients to the one presented on figure 1). The pictures presented on figure 1 are too small to evaluate if staining is localized or widespread in the glomeruli. Also, please present silver stainings and EM pictures for the same patients, especially patients with EM stage I of Churg. The point is to tease out if C1q deposition occurs early in the disease process or is a late secondary event.
2. Please provide control pronase stainings for biopsies of patients with FSGS as control in order to show that C1q and others are not non-specific staining in areas of scar or significant GBM damage.
3. This is important in view of the authors previous publication stating "Notably, MN developed in the absence of detectable complement activation, and disease was strain dependent" (Tomas et al. J Am Soc Nephrol 28: 3262-3277, 2017). This suggests that findings are not common to all MN but are strain specific and as such not generalizable.
4. I am not convinced that staining for IgG1q and IgGMBL presented in figure 2 is specific. They are certainly not of the level of IgG3 and IgG4 staining (figure 2e). Please provide larger pictures for at least 10 cases to show it is consistent
5. I am also concerned that the injection of THSD7A fragments is producing a non-specific immune response, especially since they used the BALB/c mice as this strain can produce strong humoral response. PBS-immunized controls are not the correct control. They should do the same experiments using another protein, like a GBM protein or Protocadherin-7A which is not associated with complement deposition on kidney biopsies (ref #9), to show that the activation of the 3 pathways of complement is specific to THSD7A and not part of a generalized immune response. If injection of one of these proteins does not result in MN but shows C1q etc deposition, then we know that is part of a generalized immune response.
6. Please also provide biopsy data for other tissues, e.g., lung, liver, spleen to demonstrate activation of complement is restricted to the kidney.

Minor

Line 394 - "histological features of MN including." The rest of the sentence is missing

Reviewer #2 (expertise in kidney pathology and glomerular disease):

The authors have explored complement activation in biopsies of human membranous GN and in a mouse model where membranous is induced using immunisation to an antigen involved in human

membranous GN. The generally accepted view in the literature is that complement activation in human membranous GN is predominantly via the lectin or alternative pathways of complement activation. The results here suggest that activation via the classical pathway is important/dominant in human membranous GN. As I understand it, this result is entirely dependent on the use of a single polyclonal antibody against C1q. It is not clear to me from the results what controls were used to ensure the specificity of this antibody. The claim that the classical pathway is activated is a key finding in the paper and therefore I think more details are required of controls for specificity to give confidence in this result. At the least, I think the results should be duplicated with another C1q specific antibody. In the discussion they suggest that their ability to detect C1q is because they are using paraffin sections with antigen retrieval rather than frozen sections. This is an interesting suggestion but surely an alternative explanation is that formalin fixation, paraffin embedding and antigen retrieval leads to antigen damage and non-specific antibody binding. In the second part of the study the authors established an autoimmune mouse model of membranous GN by immunisation with THSD7A. This is an important and novel model although there has been a previous description of membranous in mice induced by passive transfer of antibodies to THSD7A. They show that there is activation of the classical and alternative pathways of complement with deposition of C5b-9. Mice deficient in C3 are partly protected from disease. This is an important finding although not completely unexpected as C3 activation has been shown to be involved in other rodent models of membranous GN. They further show that inhibition of C3 synthesis using systemically administered siRNA can ameliorate established disease. This is an interesting result with implications for therapy.

Other major points

1. Why was immunofluorescence in mouse kidneys carried out subjectively when it would be easy to quantitate by image analysis measuring pixel intensity in glomeruli?

Minor points

1. Introduction, line 53. I don't think deposition of immunoglobulins and complement can be detected by light microscopy

Reviewer #3 (expertise in complement activation, innate immunity, C5 complement):

This report suggests that experimental membranous nephropathy induced in mice is triggered via complement activation involving all three pathways (classical, alternative and lectin pathways) and that all three pathways contribute to the presence of immune complexes appearing predominately in glomerular locations. The novelty of this report is the detailed morphological analysis of complexes in glomeruli, using advanced strategies that purport to quantitatively demonstrate the immune complexes. On the basis of these data, the authors suggest that the nephropathy is associated with activation of the three complement pathways, and using RNA interference strategies that "silencing" of C3 greatly reduces the intensity of the glomerular damage and that blockade of C3 might be a strategy to reduce the intensity of glomerular dysfunction in this disease.

My concerns involve several issues:

1. There is an excessive amount of images that take up a great amount of space in the figures. Obviously, the number of images could be substantially reduced.
2. The authors suggest that blockade of C3 could be considered clinically for treatment of patients. However, blocking of C3 carries with it reduced complement-related phagocytosis related to the function of both C3 and C5 that could pose a significant problem. This needs to be discussed.
3. The authors do not discuss the potential role of C5a in the disease progress and how blockade of C5a with antibody or blockade of C5a Receptor1 could be alternative strategies in membranous nephropathy.

In general, this is an interesting and useful study that needs some revision as described above.

RESPONSE TO REVIEWER COMMENTS

Reviewer #1 (expertise in nephrology, glomerular disease):

Seifert et al. evaluated complement cascade activation in a model of membranous nephropathy (MN) and found that all 3 pathways (classical, lectin, and alternative) are activated. They also found a close correlation to the degree of glomerular C1q-binding IgG subclasses. Complement-deficient mice were protected from glomerular damage. Targeting of C3 using RNA interference attenuated disease. The authors conclude that complement is primarily activated via classical pathway and that complement components such as C3 represent promising therapeutic targets in MN.

Comments:

1. The statement that C1q is present in human biopsies of patients with MN is puzzling since this reviewer has seen multiple biopsies of patients with MN and never saw C1q staining, not even when pronase has been done, except of course in cases of membranous lupus nephropathy. Similarly, mass spectrometry data does not show C1q to be present (Ref #22). Please provide additional enlarged immunofluorescence biopsy pictures of at least 10 of the human primary MN (not lupus membranous) showing C1q staining (additional patients to the one presented on figure 1). The pictures presented on figure 1 are too small to evaluate if staining is localized or widespread in the glomeruli. Also, please present silver stainings and EM pictures for the same patients, especially patients with EM stage I of Churg. The point is to tease out if C1q deposition occurs early in the disease process or is a late secondary event.

Thank you for raising this crucial point. Please note that other studies applying mass spectrometry found higher detection rates of C1q, also in primary MN cases (Ayoub et al., *Kidney Int Rep* 2021; Manral et al., *Front Immunol* 2022; notably, the latter paper used conventional IF for C1q detection with negative results and mass spectrometry with positive results for C1q).

We have now performed a multi-level validation of the C1q staining:

- **Reviewer Fig. 1** shows enlargements of the stainings that are presented in Fig. 1a-d. It can be appreciated that the C1q staining (and the stainings of the other complement components) is widespread at the subepithelial aspect of the GBM (as indicated by its location in relation to collagen IV) as well as in strong co-localization with IgG, as it would be expected in MN. We hope that these enlargements in the automatically generated PDF are of sufficient quality to properly evaluate the stainings.
- We show the C1q/IgG stainings of the remaining 4 patients from the initial pilot cohort (Fig. 1 and Supplementary Fig. 1b) in **Reviewer Fig. 2**, demonstrating that granular C1q positivity in strong co-localization with IgG is not an occasional event.
- Please note that Supplementary Table 1 shows the EM stages according to Churg and Ehrenreich for all MN cases that were investigated using proximity ligation assays (Fig. 2). There are several cases with lower stages, e.g. MN 12, MN 20, MN 21, MN 29, MN 30, MN 31, MN 32, MN 34, MN 37, MN 38. The cases with lower stages I-II according to Churg and Ehrenreich do not significantly differ in the degree of IgG/C1q positivity, suggesting that complement activation via C1q is indeed an early event in the pathogenesis of MN. This sub-analysis is shown in **Reviewer Fig. 3**.
- As suggested, we stained 10 additional cases of primary MN (8 PLA2R1-associated, 2 THSD7A-associated) for C1q using immunofluorescence and show the corresponding EM images side-by-side in **Reviewer Fig. 4**. These stainings show strong C1q positivity in both early stages and later stages according to Churg and Ehrenreich.
- In response to Reviewer 2, we have controlled the staining of the rabbit-derived anti-C1q antibody by using a rabbit IgG isotype control in the same concentration and with the same fluorochrome-labeled secondary anti-rabbit IgG antibody. This did not reveal positivity in biopsies from patients with MN, arguing against unspecific binding of the primary rabbit IgG or the secondary anti-rabbit IgG to the injured tissue/glomerular filtration barrier. This analysis is shown in **Supplementary Fig. 2**.
- In response to the request by Reviewer 2, we confirmed C1q positivity in biopsies from MN patients using another (goat-derived) anti-C1q antibody. This is shown in **Supplementary Fig. 3**. Please also see our detailed response to Reviewer 2 below.

- Interestingly, while C1q was strongly positive in the THSD7A-immunized model mice (Fig. 4e), we found C1q to be negative when IF was performed on cryo-conserved tissue of the same animals (Reviewer Fig. 5). This situation resembles the discrepancy between the literature (and your personal experience), where C1q is usually described to be negative in cases of (primary) MN when IF is performed on cryo-conserved tissue, and our study, where C1q was strongly positive in IF on paraffin-embedded tissue. We thus hypothesized, as also discussed in the previous version of the manuscript, that glomerular C1q is masked in MN. To test this, we established an antigen retrieval approach on the frozen mouse tissue using a combination of methanol and trypsin. Interestingly, this led to a granular C1q positivity in THSD7A-immunized mice, which was not present without the antigen retrieval (Reviewer Fig. 5).
- Encouraged by this finding, we set up a collaboration with the Department of Medical Genetics and Pathology, University Hospital Basel, Switzerland, where kidney biopsies are routinely partly paraffin-embedded and partly cryo-conserved. We stained frozen sections (6 cases of PLA2R1-associated MN, 2 cases of class V lupus nephritis, and 2 cases of minimal change disease) for C1q using both conventional indirect IF without tissue preprocessing and our custom-made protocol involving antigen retrieval, which was previously established in mice. While C1q was positive with and without antigen retrieval in cases of lupus nephritis, C1q was positive in cases of PLA2R1-associated MN only after antigen retrieval (Fig. 1e). We did not detect this signal when an isotype control (rabbit IgG) was used instead of the rabbit anti-C1q antibody in the same concentration and followed by the same fluorochrome-labeled secondary antibody (Supplementary Fig. 5). Additionally, the C1q positivity could be reproduced with a second (goat-derived) anti-C1q antibody (Supplementary Fig. 6), and cases of minimal change disease were negative for C1q with both protocols and C1q antibodies (Fig. 1e and Supplementary Fig. 6).

In summary, we hope that our experiments convincingly demonstrate the presence of C1q in kidney biopsies from patients with (primary) MN.

2. Please provide control pronase stainings for biopsies of patients with FSGS as control in order to show that C1q and others are not non-specific staining in areas of scar or significant GBM damage.

Thank you for this comment. We now show C1q stainings for biopsies from 5 patients with primary FSGS, 5 patients with minimal change disease, 5 patients with diabetic nephropathy, and 5 patients with amyloidosis. No relevant positivity of C1q could be detected at the glomerular filtration barrier in these cases. These images are shown in Supplementary Fig. 4.

3. This is important in view of the authors previous publication stating “Notably, MN developed in the absence of detectable complement activation, and disease was strain dependent” (Tomas et al. J Am Soc Nephrol 28: 3262–3277, 2017). This suggests that findings are not common to all MN but are strain specific and as such not generalizable.

Thank you for bringing up this important point. Please note that the cited paper investigated the sequelae of a passive transfer of rabbit anti-THSD7A antibodies. We do not know the capacity of rabbit anti-THSD7A IgG binding to mouse C1q required for subsequent complement activation. From the results of the mentioned study we may speculate that this capacity is, at best, limited. It is correct that mice develop proteinuria after transfer of rabbit anti-THSD7A antibodies, also without significant activation of complement. This is in accordance with the results of the present study, where C3-deficient mice also developed proteinuria upon immunization with THSD7A, yet substantially less than complement-competent mice. The limitations of passive transfer models for investigating MN pathophysiology are discussed in detail on the second page of the discussion.

4. I am not convinced that staining for IgGC1q and IgGMBL presented in figure 2 is specific. They are certainly not of the level of IgG3 and IgG4 staining (figure 2e). Please provide larger pictures for at least 10 cases to show it is consistent

Thank you for raising this important matter. Please note that we do not necessarily expect the proximity ligation assay signals to be of the same level/intensity as the immunostainings for IgG subclasses or single complement components. The proximity ligation assay only gives a signal if the two components (e.g. IgG and C1q or IgG and MBL) are located in direct proximity (around 30-40 nm), allowing conclusions on complement activity in the area of a positive signal. In the context of a running complement cascade, where molecules are cleaved, bind, and dissociate to and from each other, we do not necessarily expect every IgG molecule to be bound to C1q, as we do not necessarily expect

every C2b molecule to be bound to C4b (classical/lectin convertase) or C3b to be bound to Bb (alternative convertase) etc. The proximity ligation assay rather detects the extent of complement activation through the corresponding pathway. We believe that the main advantage of the proximity ligation assay is the possibility to objectively quantify molecules in direct proximity, allowing conclusions on the degree of complement activation through the different pathways rather than an accumulation/deposition of complement factors only.

We show images from one representative glomerulus from all biopsy samples that were investigated by using proximity ligation assays for IgGC1q ($n=39$) and IgGMBL ($n=26$) in Reviewer Fig. 6 and Reviewer Fig. 7, respectively.

5. I am also concerned that the injection of THSD7A fragments is producing a non-specific immune response, especially since they used the BALB/c mice as this strain can produce strong humoral response. PBS-immunized controls are not the correct control. They should do the same experiments using another protein, like a GBM protein or Protocadherin-7A which is not associated with complement deposition on kidney biopsies (ref #9), to show that the activation of the 3 pathways of complement is specific to THSD7A and not part of a generalized immune response. If injection of one of these proteins does not result in MN but shows C1q etc deposition, then we know that is part of a generalized immune response.

Thank you for this comment. We fully agree that differentiating a systemic, generalized immune response from local antibody action is highly important in this context. Several autoimmune models rely on active immunization with an antigen emulsified in an adjuvant to enhance the immune response and thus (auto)antibody generation. These models include, for example, active Heymann nephritis (AHN, a model of MN), experimental autoimmune glomerulonephritis (EAG, a model of crescentic glomerulonephritis), and experimental autoimmune encephalomyelitis (EAE, a model of multiple sclerosis). Complete Freund's adjuvant (CFA), which is usually used for the first immunization, contains inactive Mycobacteria, which enhance the immune response. As CFA contains bacterial proteins, this immunization technique always involves immunization with antigens different from the antigen of interest. Thus, as done in our experiments, the use of PBS in place of the antigen of interest, but in combination with CFA, involves immune stimulation as well as immunization with irrelevant proteins and adequately controls for a non-specific immune response. The above-mentioned models usually use PBS/CFA, CFA alone, or untreated animals as controls. Please refer to, for example, Kerjaschki and Farquhar, PNAS 1982 (AHN), Hopfer et al., J Immun 2015 (EAG), Olechowski et al., Experimental Neurology 2009 (EAE), Prinz et al., PLoS One 2015 (EAE).

However, we understand your concern and feel that the unequivocal demonstration that the observed effects are a consequence of local autoantibody action and not a consequence of a generalized immune response is crucial. To address this issue, we have performed the following additional experiments:

1. We immunized WT mice with ovalbumin/CFA/IFA using the established immunization protocol in comparison with our prior approach involving PBS/CFA/IFA. Ovalbumin is often used in immunological studies to control for non-specific systemic effects. Ovalbumin-immunized animals developed anti-ovalbumin antibodies, but no albuminuria and no histological signs of MN such as glomerular IgG or complement deposition, indicating that the observed changes in THSD7A-immunized mice are not a consequence of a generalized immune response. These results are presented in Supplemental Fig. 12.
2. We immunized conditional *Thsd7a*^{-/-} mice and their WT littermates with THSD7A using the established immunization protocol. While WT littermates, as expected, developed anti-THSD7A autoantibodies, proteinuria and the morphological signs of MN including glomerular complement deposition, *Thsd7a*^{-/-} mice were completely protected from clinical and histological MN – despite developing high levels of anti-THSD7A antibodies (as expected in the context of genetic THSD7A deficiency). This demonstrates that the clinical and morphological development of MN, including glomerular complement deposition, is a consequence of local THSD7A-anti-THSD7A interaction and excludes disease induction by systemic effects or antibodies different from anti-THSD7A. We show these results in Supplemental Fig. 13.
3. To further investigate whether MN development is a direct consequence of local anti-THSD7A antibody binding or a consequence of unspecific systemic effects in the context of immunization, we purified mIgG from the diseased THSD7A-immunized WT animals (Fig. 3) and passively transferred it to a *Thsd7a*^{-/-} mouse and a WT littermate of the *Thsd7a*^{-/-} mouse. The WT littermate, but not the *Thsd7a*^{-/-} mouse, developed granular deposition of mIgG, C1q and C3 in immunofluorescence as well as subepithelial electron-dense deposits and podocyte foot process effacement in electron microscopy. Importantly, the WT littermate, but not the *Thsd7a*^{-/-} mouse, developed high albuminuria, and anti-THSD7A antibodies could only be eluted from

kidney sections of the WT littermate. This demonstrates that the development of MN with glomerular complement activation is a consequence of local anti-THSD7A antibody binding. These experiments are shown in Reviewer Fig. 8.

6. Please also provide biopsy data for other tissues, e.g., lung, liver, spleen to demonstrate activation of complement is restricted to the kidney.

We have now taken several additional organs from THSD7A-immunized mice and controls (lung, spleen, liver, heart, testicle) and stained for mIgG, C1q and C3. We found no relevant deposition of these molecules in organs other than the kidney, arguing against non-specific systemic complement activation in the context of active immunization. This data is shown in Supplementary Fig. 15.

Minor

Line 394 – “histological features of MN including.” The rest of the sentence is missing

The word “including” was removed.

Reviewer #2 (expertise in kidney pathology and glomerular disease):

The authors have explored complement activation in biopsies of human membranous GN and in a mouse model where membranous is induced using immunisation to an antigen involved in human membranous GN. The generally accepted view in the literature is that complement activation in human membranous GN is predominantly via the lectin or alternative pathways of complement activation. The results here suggest that activation via the classical pathway is important/dominant in human membranous GN. As I understand it, this result is entirely dependent on the use of a single polyclonal antibody against C1q. It is not clear to me from the results what controls were used to ensure the specificity of this antibody. The claim that the classical pathway is activated is a key finding in the paper and therefore I think more details are required of controls for specificity to give confidence in this result. At the least, I think the results should be duplicated with another C1q specific antibody. In the discussion they suggest that their ability to detect C1q is because they are using paraffin sections with antigen retrieval rather than frozen sections. This is an interesting suggestion but surely an alternative explanation is that formalin fixation, paraffin embedding and antigen retrieval leads to antigen damage and non-specific antibody binding.

Thank you for bringing up these essential points. Please also see our comments to Reviewer 1 above. We have now performed additional controls and a validation with another anti-C1q antibody. This is a goat-derived antibody raised against human C1q (Complement Technologies), which cross-reacts with murine C1q, and is the antibody that we used to detect C1q in our model mice (e.g. Fig. 4e).

We have stained 5 biopsies (paraffin-embedded tissue) from patients with primary MN as well as control biopsies from renal transplant recipients, primary FSGS, minimal change disease, diabetic nephropathy and amyloidosis using this antibody. This experiment revealed C1q positivity in biopsies from MN patients, while control biopsies were negative for C1q at the glomerular filtration barrier. These results are shown in **Supplementary Fig. 4**.

In addition, we performed stainings on biopsies from MN patients (paraffin-embedded tissue) using a rabbit isotype control IgG instead of the rabbit anti-C1q antibody at an equal concentration, followed by the identical fluorochrome-conjugated anti-rabbit IgG secondary antibody, to control for unspecific binding of rabbit IgG or the secondary anti-rabbit IgG antibody to the injured tissue/glomerular filtration barrier. These experiments did not reveal significant positivity, excluding that the C1q signal that we found in biopsy specimens from patients with MN is an artifactual consequence of formalin fixation, paraffin embedding or antigen retrieval in the context of an injured glomerular filtration barrier. The results of two representative stainings are shown in **Supplementary Fig. 2**.

Finally, we stained C1q in cases of PLA2R1-associated MN (n=6), class V membranous lupus nephritis (n=2), and minimal change disease (n=2) on frozen biopsy samples using both conventional indirect IF without tissue preprocessing and a custom-made protocol involving antigen retrieval with methanol and trypsin. While C1q was positive with and without antigen retrieval in cases of lupus nephritis, C1q was positive in cases of PLA2R1-associated MN only after antigen retrieval (**Fig. 1e**). Importantly, we did not detect this signal when an isotype control (rabbit IgG) was used on place of the rabbit anti-C1q antibody in the same concentration and followed by the same fluorochrome-labeled secondary antibody (**Supplementary Fig. 5**). Moreover, C1q positivity in MN could be reproduced with a second (goat-derived) anti-C1q antibody (**Supplementary Fig. 6**) and cases of minimal change disease were negative for C1q with both protocols.

In the second part of the study the authors established an autoimmune mouse model of membranous GN by immunisation with THSD7A. This is an important and novel model although there has been a previous description of membranous in mice induced by passive transfer of antibodies to THSD7A. They show that there is activation of the classical and alternative pathways of complement with deposition of C5b-9. Mice deficient in C3 are partly protected from disease. This is an important finding although not completely unexpected as C3 activation has been shown to be involved in other rodent models of membranous GN. They further show that inhibition of C3 synthesis using systemically administered siRNA can ameliorate established disease. This is an interesting result with implications for therapy.

Thank you for these encouraging comments. We hope that the potential of the easy-to-apply “autoimmune” model presented in this study, involving both antibody-producing cells as well as specific autoantibodies and their local action on podocytes, to investigate different additional scientific questions will be appreciated.

Other major points

1. Why was immunofluorescence in mouse kidneys carried out subjectively when it would be easy to quantitate by image analysis measuring pixel intensity in glomeruli?

Thank you for bringing this up. When staining for mIgG subclasses in murine kidney samples, mesangial positivity could regularly be observed in a subset of glomeruli (despite perfusion through the heart when sacrificing the animals to minimize vascular mIgG positivity). **Reviewer Fig. 9** shows examples of mesangial positivity for mIgG2a in two control animals and two THSD7A-immunized animals. Similarly, C5b-9 positivity can also sometimes be appreciated in a mesangial, intravascular or, particularly in diseased animals, intracellular localization (please see Supplementary Fig. 14b and Supplementary Fig. 16g, and please refer to Kerjaschki *et al.*, Transcellular transport and membrane insertion of the C5b-9 membrane attack complex of complement by glomerular epithelial cells in experimental membranous nephropathy, *J Immun* 1989). As such positivity would confound a fluorescence intensity analysis, we considered a semiquantitative scoring, which only takes into account positivity at the glomerular filtration barrier, as the more valid method to evaluate the degree of mIgG and C5b-9 deposition. Please note that these analyses were performed in a blinded manner, as described in detail the methods. These aspects do not apply for nephrin, neph1 and synaptopodin stainings. We have thus re-analyzed the images from Fig. 6 using ImageJ – with very similar results when compared with the previous semiquantitative analysis. We have now replaced the semiquantitative analyses in **Figure 6d-f** with the more objective quantitative pixel analysis, as suggested.

Minor points

1. Introduction, line 53. I don't think deposition of immunoglobulins and complement can be detected by light microscopy

We have deleted "or light microscopy".

Reviewer #3 (expertise in complement activation, innate immunity, C5 complement):

This report suggests that experimental membranous nephropathy induced in mice is triggered via complement activation involving all three pathways (classical, alternative and lectin pathways) and that all three pathways contribute to the presence of immune complexes appearing predominately in glomerular locations. The novelty of this report is the detailed morphological analysis of complexes in glomeruli, using advanced strategies that proopt to quantitatively demonstrate the immune complexes. On the basis of these data, the authors suggest that the nephropathy is associated with activation of the three complement pathways, and using RNA interference strategies that “silencing” of C3 greatly reduces the intensity of the glomerular damage and that blockade of C3 might be a strategy to reduce the intensity of glomerular dysfunction in this disease.

My concerns involve several issues:

1. There is an excessive amount of images that take up a great amount of space in the figures. Obviously, the number of images could be substantially reduced.

Thank you for this comment. We understand your concern and have moved images from former Fig. 5h to the Supplement. Overall, we strongly feel that the main messages of this manuscript rely on the convincing presentation of histological images.

2. The authors suggest that blockade of C3 could be considered clinically for treatment of patients. However, blocking of C3 carries with it reduced complement-related phagocytosis related to the function of both C3 and C5 that could pose a significant problem. This needs to be discussed.

This is an important point. We have added this aspect to the discussion.

3. The authors do not discuss the potential role of C5a in the disease progress and how blockade of C5a with antibody or blockade of C5a Recepto1 could be alternative strategies in membranous nephropathy.

In general, this is an interesting and useful study that needs some revision as described above.

Thank you for this comment. We now discuss this point in the last paragraph of the discussion.

Reviewer Figure 1

a

b

c

d

Reviewer Figure 1 Enlargements of the stainings from main Fig. 1. a-d Representative immunofluorescence stainings for C3b and CFB (a), C4b and C2 (b), C1q and IgG (c), and MBL and IgG (d) in co-localization with the glomerular basement membrane constituent collagen IV in biopsies from MN patients and controls. The lower panels represent 5-fold enlargements of the boxed areas in the upper panels. Bars 50 μ m.

Reviewer Figure 2

PLA2R1-associated MN case 1

PLA2R1-associated MN case 2

PLA2R1-associated MN case 3

THSD7A-associated MN case 1

Reviewer Figure 2 C1q and IgG co-localization in 4 additional biopsy samples from patients with MN. C1q and IgG staining in co-localization with the glomerular basement membrane constituent collagen IV in biopsies from MN patients. The lower panels represent 5-fold enlargements of the boxed areas in the upper panels. Bars 50 μ m. THSD7A-associated MN case 2 is shown in Fig. 1c.

Reviewer Figure 3

Reviewer Figure 3 IgGC1q proximity ligation assay signals in samples with low and high EM stages according to Churg. The IgGC1q proximity ligation assay signals did not differ significantly between biopsy samples from MN patients with a lower stage (stage I-II according to Churg) and samples with a higher stage (stage III-IV according to Churg), suggesting that complement activation via C1q fixation is not a late secondary event in the pathogenesis of MN. EM stages from all analyzed samples are shown in Supplemental Table 1.

Reviewer Figure 4

PLA2R1 case A
EM stage: 2

PLA2R1 case B
EM stage: 1-2

Reviewer Figure 4 (continued)

PLA2R1 case C
EM stage: 2

PLA2R1 case D
EM stage: 1-2

Reviewer Figure 4 (continued)

PLA2R1 case E
EM stage: 3

PLA2R1 case F
EM stage: 3

Reviewer Figure 4 (continued)

PLA2R1 case G
EM stage: 1

PLA2R1 case H
EM stage: 1

Reviewer Figure 4 (continued)

THSD7A case A
EM stage: 3

THSD7A case B
EM stage: 1

Reviewer Figure 4 C1q positivity in correlation with EM stages according to Churg. C1q staining in co-localization with collagen IV and electron microscopic images in 8 cases of PLA2R1-associated MN and 2 cases of THSD7A-associated MN of different electron microscopy (EM) stages according to Churg. The one C1q-negative case (PLA2R1 case A) had stage 2 MN according to Churg. Bars 20 μ m.

Reviewer Figure 5

Reviewer Figure 5 C1q becomes detectable in frozen kidney sections of THSD7A-immunized mice after antigen retrieval. **a** Immunofluorescence (IF) staining (paraffin-embedded tissue) for C1q in co-localization with wheat germ agglutinin (WGA) in a THSD7A-immunized mouse. **b** IF staining (cryoconserved tissue) for C1q in co-localization with laminin and endomucin using conventional indirect IF (upper panels) and IF after antigen retrieval with trypsin and methanol (lower panels) in the same THSD7A-immunized mouse. Bars 20 μ m.

Reviewer Figure 6

MN 1

MN 2

MN 3

MN 4

MN 5

MN 6

MN 7

MN 8

MN 9

MN 10

MN 11

MN 12

Reviewer Figure 6 (continued)

MN 13

MN 14

MN 15

MN 16

MN 17

MN 18

MN 19

MN 20

MN 21

MN 22

MN 23

MN 24

Reviewer Figure 6 (continued)

MN 25

MN 26

MN 27

MN 28

MN 29

MN 30

MN 31

MN 32

MN 33

MN 34

MN 35

MN 36

Reviewer Figure 6 (continued)

Reviewer Figure 6 Representative IgGC1q proximity ligation assay images from all investigated biopsies from MN patients and controls. Lower images are enlargements of the boxed areas in the upper images. Scale bar in the first image applies to all upper images and indicates 50 μ m.

Reviewer Figure 7

Reviewer Figure 7 (continued)

MN 13

MN 14

MN 15

MN 16

MN 17

MN 18

MN 19

MN 20

MN 22

MN 27

MN 29

MN 30

Reviewer Figure 7 (continued)

Reviewer Figure 7 Representative IgGMBL proximity ligation assay images from all investigated biopsies from MN patients and controls. Lower images are enlargements of the boxed areas in the upper images. Scale bar in the first image applies to all upper images and indicates 50 μ m.

Reviewer Fig. 8

Reviewer Fig. 8 IgG transfer from diseased THSD7A-immunized wild-type (WT) mice to a *Thsd7a^{-/-}* mouse and a WT littermate. **a** Experimental setup: Mouse IgG (mlgG) was purified from diseased THSD7A-immunized WT mice and 3 mg of purified mlgG were transferred to one *Thsd7a^{-/-}* mouse and one WT littermate. **b** Proteinuria as measured by albumin-to-creatinine ratio in the *Thsd7a^{-/-}* mouse and the WT littermate after mlgG transfer. **c, d** Immunofluorescence analyses of the *Thsd7a^{-/-}* mouse and the WT littermate: stainings for mlgG and THSD7A in colocalization with nephrin (**c**), and complement C1q and C3 in co-localization with WGA (**d**). Bars 20 μ m. **e** Electron microscopic analysis of the *Thsd7a^{-/-}* mouse and the WT littermate after mlgG transfer showing podocyte foot process effacement in the WT littermate. Bars 2 μ m. **f** Western blot analysis of antibodies eluted from frozen kidney sections of the *Thsd7a^{-/-}* mouse and the WT littermate on recombinant THSD7A and mouse glomerular extracts (MGE).

Reviewer Fig. 9

Reviewer Fig. 9 Examples of mesangial mIgG positivity. Immunofluorescence analysis of mIgG2a in two control mice and two THSD7A-immunized mice showing variable mesangial positivity for mIgG2a independent of the positivity at the filtration barrier. Bars 20 μ m.

REVIEWERS' COMMENTS

Reviewer #1 (expert in nephrology and glomerular disease):

The authors have addressed my comments in detail.

I would like to point out however that the paper by Ayoub et al (KIR) mentioned in the reply, in fact does not show C1q. The spectra count is equal or less than baseline PLA2R counts and 100 times less than C3. As such, it is background, baseline readings, and possible due to serum contamination of the biopsy specimen.

Reviewer #2 (expert in kidney histopathology and glomerular disease):

The authors have carefully considered the comments in my original review and have carried out new experiments to answer my concerns

Reviewer #3 (expert in complement activation and innate immunity):

I believe the authors have responded adequately to all three reviewers and have answered concerns of each of the three reviewers. The information adds important information related to immune responses that cause membranous nephropathy which is an important cause for this type of nephropathy.